# Atypical chemoreceptor arrays accommodate high membrane curvature

Alise R. Muok [1,2], Davi R. Ortega [3], Kurni Kurniyati[4], Wen Yang[1,2], Zachary A. Maschmann [5], Adam Sidi Mabrouk [1,2], Chunhao Li [4], Brian R. Crane [5] & Ariane Briegel [1,2✉]

The prokaryotic chemotaxis system is arguably the best-understood signaling pathway in biology. In all previously described species, chemoreceptors organize into a hexagonal (P6 symmetry) extended array. Here, we report an alternative symmetry (P2) of the chemotaxis apparatus that emerges from a strict linear organization of the histidine kinase CheA in Treponema denticola cells, which possesses arrays with the highest native curvature investigated thus far. Using cryo-ET, we reveal that Td chemoreceptor arrays assume an unusual arrangement of the supra-molecular protein assembly that has likely evolved to accommodate the high membrane curvature. The arrays have several atypical features, such as an extended dimerization domain of CheA and a variant CheW-CheR-like fusion protein that is critical for maintaining an ordered chemosensory apparatus. Furthermore, the previously characterized Td oxygen sensor ODP influences CheA ordering. These results suggest a greater diversity of the chemotaxis signaling system than previously thought.

[1] Institute for Biology, Leiden University, Sylviusweg 72, 2333 BE Leiden, Netherlands. [2] Centre for Microbial Cell Biology, Leiden University, Sylviusweg 72, 2333 BE Leiden, Netherlands. [3] Department of Biology, California Institute of Technology, 1200 E. California Blvd., Pasadena, CA 91125, USA. [4] Department of Oral and Craniofacial Molecular Biology, Philips Research Institute for Oral Health, Virginia Commonwealth University, Richmond, VA 23298, USA. [5] Department of Chemistry and Chemical Biology, Cornell University, Ithaca, NY 14850, USA. ✉email: a.briegel@biology.leidenuniv.nl

Chemotaxis is a behavior most motile bacteria employ to sense their chemical environment and navigate toward favorable conditions. The main components of the system are transmembrane chemotaxis receptors called methyl-accepting chemotaxis proteins (MCPs), the histidine kinase CheA, and the adapter protein CheW. The intracellular tips of MCPs bind CheA and CheW, and communicate changes from the external chemical environment into the cell by modulating CheA kinase activity (Fig. 1a)[1–3]. Activation of CheA initiates an intracellular phosphorelay that ultimately controls flagellar rotation and cell movement. CheA functions as a dimer and possesses five domains (P1-P5) with distinct roles in autophosphorylation and array integration. The P1 domain is the phosphate substrate domain, P2 interacts with response regulators, P3 is the dimerization domain, P4 binds ATP, and P5 interacts with CheW. In the model species *Escherichia coli* (*Ec*), CheA P5 and CheW are paralogs that interact pseudo-symmetrically to form six-subunit rings. In all bacterial and archaeal species examined thus far, the MCPs are arranged in a trimer-of-dimer oligomeric state and further

organize into a hexagonal lattice (Fig. 1b, c). In *Ec*, the receptors are connected by the highly ordered rings of CheA and CheW bound to the cytoplasmic tips of the receptors (Fig. 1d)[4,5]. These insights have established a widely accepted central model of the chemotaxis array (Fig. 1a–d)[4,6,7]. However, emerging research has recently revealed divergent components and arrangements of the chemotaxis apparatus in non-canonical organisms. For example, in *Vibrio cholerae* (*Vc*) chemotaxis arrays, CheA and CheW lack an ordered arrangement in the rings[8,9]. Many of these structural insights have transpired from cryo-electron tomography (cryo-ET) studies that utilize artificial systems for higher resolution data. Specifically, the advent of so-called "mini-cell" bacterial strains produce extremely small cells that are ideal for cryo-ET[8,10], and lipid-templating methods generate in vitro arrays with increased conformational homogeneity[11,12]. However, these methods generate arrays with non-native curvature, and it is unclear how this may affect array structure and behavior.

Here, we used cryo-ET to examine the in vivo array structure of the pathogenic spirochete *Treponema denticola* (*Td*), which

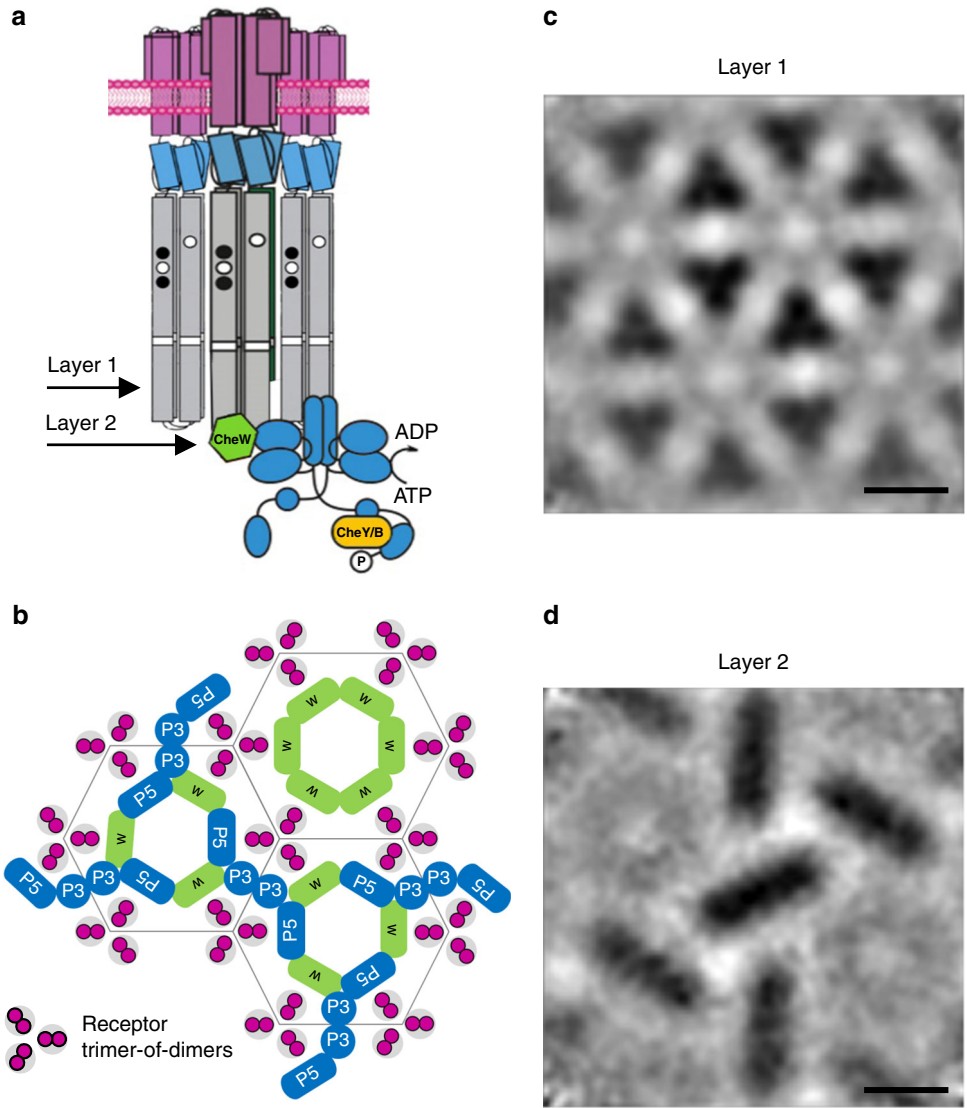

**Fig. 1 The arrangement of the canonical chemoreceptor array, exemplified in *Escherichia coli* (*Ec*). a** Transmembrane chemoreceptors function as a trimer-of-dimers module to modulate the activity of the histidine kinase CheA (blue) via a coupling protein CheW (green). In cryo-ET experiments, cross-sections at Layer 1 reveal the arrangement of chemoreceptors and Layer 2 reveals the position of the kinase. **b** In all organisms investigated previously, CheA is integrated into the array with hexagonal (pseudo-P6) symmetry. **c** Previous cryo-ET experiments in *E. coli* reveal the hexagonal receptor arrangement at Layer 1 and **d** the pseudo-P6 CheA arrangement at Layer 2 (EMDB-4991)[25]. Scale bars are 6 nm.

possesses the highest native membrane curvature of any bacterial species examined for its chemotaxis system so far[4,7]. We demonstrate the presence of an array architecture with two-fold (P2) symmetry in *Td*, which is likely caused by the high curvature of the cells. Genetic experiments, bioinformatics analyses, structural investigations, and molecular modeling of *Td* chemotaxis proteins reveal adaptations that have likely evolved to accommodate formation of an extended chemotaxis array in a highly curved membrane. We demonstrate that a CheR-like fusion domain in a variant CheW is key for maintaining the structural integrity of the arrays. Furthermore, cryo-ET analysis of *Td* cells lacking the oxygen sensor ODP reveals substantial changes in the ordering or mobility of CheA[13]. Collectively, these data demonstrate a greater diversity of the chemotaxis system than previously realized and exemplify the importance of examining biological structures in native in vivo conditions.

## Results

**Conservation of the F2 chemotaxis system.** The chemotaxis systems in prokaryotes have been classified into 19 systems based on phylogenomic markers[14]. These classes include 17 systems predicted to control "Flagellar motility" (F1-17), one "Alternative Cellular Function" system (ACF), and one "Type Four Pilus system" (TFP). The spirochete chemotaxis system belongs to the F2 category, which has not been investigated with structural methods[14]. To explore the characteristics of chemotaxis proteins in F2 genomes, we analyzed genomes in the Microbial Signal Transduction Database version 3 (MiST3)[15]. All genomes (306) with at least one CheA of the F2 class (CheA-F2) are in the Spirochaetota phylum, with a few exceptions of lateral gene transfer to other phyla (Dataset 1, see Methods). However, not all Spirochaetota bacteria possess an F2 system. There are 1096 genomes assigned to the Spirochaetota phylum in the Genome Taxonomy Database (GTDB)[16], a microbial taxonomy based on genome phylogeny, and 804 of these are present in the MiST3 database (Dataset 2)[15]. The GTDB taxonomy tree has 235 representative species, and 115 are in MiST3. Within these 115 genomes, we mapped the different classes of CheA kinases to the Spirochaetota taxonomy tree (Fig. S1). Based on the topology of this tree, it appears that the major chemosensory systems in the genomes from the Spirochaetota phylum are: F1/F8 (Leptospirae), F7/F2 (Brachyspirae), and F2 (Spirochaetia; Fig. 2). Interestingly, as the Brachyspirae appear between Leptospirae and Spirochaetia, we found that their F2 systems have elements of the F1 system, perhaps a transitional hybrid F1/F2 system (Dataset 2). Therefore, we conclude that complete F2 systems are exclusive to the Spirochaetia class, with a few exceptions of lateral gene transfer.

The main architectural difference of the F2 system compared to others is the presence of an unusual scaffold protein that consists of an N-terminal CheW domain and a C-terminal CheR-like domain, hereafter referred to as CheW-CheR$_{like}$. Typically, CheR is a methyltransferase that, together with the methylesterase CheB, controls the methylation state of the receptors and thus provides an adaptation system[17]. Our analyses indicate that all Spirochaetia genomes in MiST3 contain CheW-CheR$_{like}$. Furthermore, if we limit our analysis to genomes that are fully sequenced (see Methods section), CheW-CheR$_{like}$ is found only in F2 chemosensory systems.

To investigate sequence patterns in the CheR protein and the CheR-like domain of CheW-CheR$_{like}$, we produced a sequence dataset with 83 CheR-F2 and 88 CheW-CheR$_{like}$ proteins and summarized them in sequence logos (Fig. S2a). Although the key catalytic residues are conserved in the typical CheR protein, two residues that are essential for CheR to methylate chemoreceptors, R79 and Y218 in *Td* CheR, are modified in the CheW-CheR$_{like}$

protein (R79W and Y218F)[17]. Furthermore, the conserved region at the C-terminus of CheR is not conserved in CheW-CheR$_{like}$. Based on these results, we speculate that the CheR$_{like}$ domain does not possess methyltransferase activity. Collectively, our analyses suggest that CheR and CheW-CheR$_{like}$ have different biological functions.

F2 systems contain three proteins with a CheW domain: the classical scaffold CheW, CheW-CheR$_{like}$, and the histidine kinase CheA. To investigate sequence patterns of the three CheW domains, we analyzed non-redundant sequence datasets of CheW-F2, CheW-CheR$_{like}$, and CheA-F2 from all 117 genomes with at least one CheA-F2. The final alignments for each group contain the CheW domain portion of 74 CheW proteins, 59 CheW-CheR$_{like}$ proteins, and 73 CheA-F2 proteins. The sequences of each group are summarized in sequence logos and demonstrate the presence of conserved regions at established interaction interfaces, as well as loop insertions near these interfaces that could confer altered specificity of binding (Fig. S2b).

**The structure of the *Treponema denticola* (*Td*) chemotaxis array in wild-type cells.** Cell poles of intact *Td* cells were imaged by cryo-electron tomography (cryo-ET) and used for three-dimensional reconstructions. Top views (cross-sections through the array) and side views (visualizing the long axes of the receptors) of membrane-associated arrays were clearly visible (Figs. 3a and S4a). Sub-tomogram averaging revealed the conserved receptor trimer-of-dimers in the typical hexagonal arrangement (EMD-11385). Remarkably, several novel features of the chemotaxis arrays are apparent (Fig. 3B). Specifically, a density of unknown origin is located in the center of the receptor hexagons and slightly above the plane of the CheA:CheW rings. This density, which will hereafter be referred to as the "middle density", extends from two subunits in the rings (Fig. 4a). Additionally, there are small but distinct puncta of density in between some of the trimer-of-dimer modules (Fig. 3b). However, averages of the arrays at the CheA:CheW layer did not reveal discernible CheA density, indicating either a sparse or disordered distribution of CheA or a highly mobile kinase (Fig. 3c). Remarkably, the sub-tomogram averages reveal the axis of the *Td* cells relative to the chemotaxis arrays, demonstrating that the arrays occupy a preferred orientation with respect to the cell axis (Fig. S4b).

**Arrays in *T. denticola* deletion mutants.** The oxygen-binding diiron protein (ODP) functions as an oxygen sensor for chemotaxis in *Td*. However, it is unknown whether ODP is an integral component of the array[13]. This protein is genetically coupled to an MCP homolog, TDE2496, that lacks both transmembrane and sensing modules but is capable of modulating CheA activity[13]. TDE2496 likely integrates into the cytoplasmic regions of the membrane-bound arrays based on the observation that no cytoplasmic arrays were observed in the tomograms. Moreover, the *Td* genome encodes only one CheA homolog, and cytoplasmic receptors often associate with distinct kinases[5,18]. To determine if the presence of ODP (TDE2498) and its cognate receptor (TDE2496) impacts array architecture or integrity, we conducted cryo-ET with two *Td* gene knock-out strains, Δ2498 and Δ2498Δ2496[13]. Importantly, deletion of ODP does not impact transcription of TDE2496[13]. The sub-tomogram averages of these strains reveal distinct differences in array densities compared to the wild-type (WT) strain (EMD-11381 and EMD-11384; Figs. 3b, c and S4a). Namely, the location of CheA at the CheA:CheW layer (Layer 2, Fig. 3c) is now clearly visible. Interestingly, CheA arranges in well-ordered linear rows. Placement of CheA

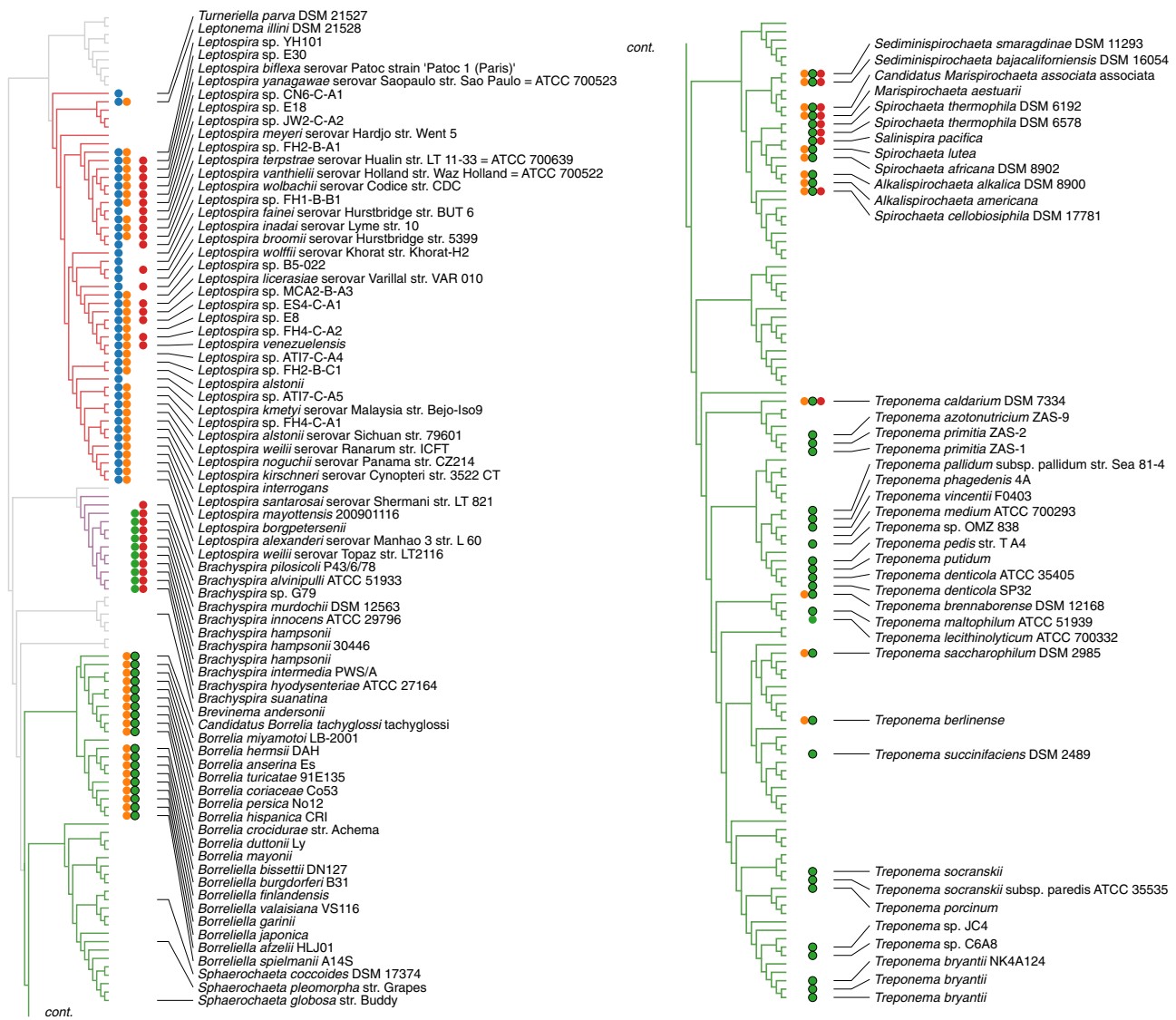

**Fig. 2 The profile of most common chemosensory classes in Spirochaetota.** The three major taxonomy classes are colored: Leptospirae (red), Brachyspirae (purple), and Spirochaetia (green). Species found in MiST3 are represented by their species and strain name. The profile shows the presence of the common chemotaxis classes F1(blue), F8(orange), F2 (green), and F7(red). The systems with CheW-CheR$_{like}$ (F2 systems) are marked with a black outline. An additional high-resolution figure is separately available.

necessarily positions the P3 domain in between two of the trimer-of-dimer modules in each hexagon. This position exclusively corresponds to the location of the puncta between receptor trimer-of-dimer modules observed in the WT array, indicating that this density represents the P3 domain (Layer 1, Fig. 3b). Like the WT arrays, the cell axis relative to the chemotaxis arrays is also apparent in these sub-tomogram averages and matches the preferred array orientation in the WT cells (Fig. S4b).

**Analyses of the CheW-CheR$_{like}$ protein in _T. denticola_.** As shown above, CheW-CheR$_{like}$ is a conserved component of the F2 Spirochaetia chemotaxis system (Figs. 2, S1). In _Td_, a 28-residue linker in CheW-CheR$_{like}$ separates the two domains and is predicted to be largely helical (Fig. S5b). The gene is co-transcribed with the only _cheA_ in the genome[18]. Upon purification, the purified protein, which is primarily a monomer with a minor dimeric component (Fig. S5a), binds strongly to CheA (Fig. 4b). The CheR$_{like}$ domain has only ~20% identity to the classical _Td_ CheR methyltransferase (L-align), but native mass spectrometry and isothermal calorimetry experiments demonstrate that CheR$_{like}$

binds the substrate S-adenosylmethionine (SAM) in a 1:1 ratio with comparatively higher affinity ($K_d = 8.8 \pm 1.9\ \mu M$) than the classical CheR ($K_d = 20.0 \pm 3.7\ \mu M$; Figs. 4c and S5c). However, the CheR$_{like}$ domain does not possess the two strictly conserved residues that are essential for methyltransferase activity[17] (R79 and Y218 in _Td_ classical CheR) or the two strictly conserved C-terminal sub-domain residues responsible for binding receptors (Gly 152 and Val180 in _Td_ classical CheR)[19]. To examine patterns of residue conservation of the folded proteins, homology models of CheR$_{like}$ and CheR were generated using a crystal structure of the classical CheR from _Salmonella typhimurium_ (PDB ID: 1AF7)[20], and the sequence logos were mapped onto the homology models (Consurf). These models reveal that residues adjacent to the SAM pocket and residues on the sub-domain that typically interact with receptors are significantly less conserved in CheR$_{like}$ (Fig. 4d). Likewise, the CheR$_{like}$ domain is more conserved on the surface opposing the SAM pocket, indicating that this surface may be important for function (Fig. 4d). Collectively, these results suggest that the CheR$_{like}$ domain has a different biological function than the classical _Td_ CheR homolog.

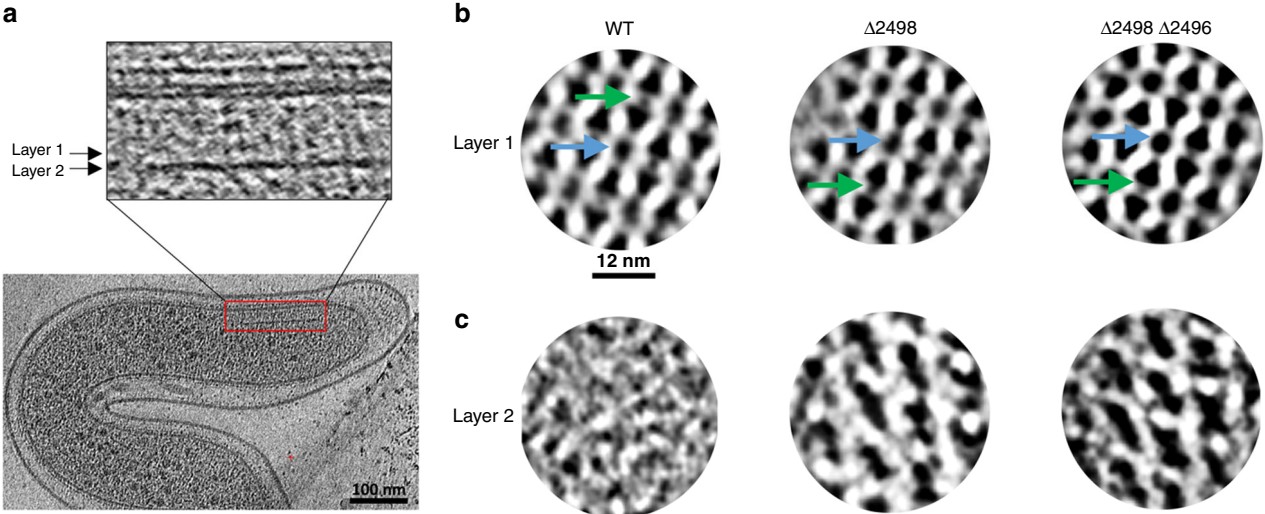

**Fig. 3 Cryo-electron tomography of whole *T. denticola* (*Td*) cells reveals the protein arrangement of chemotaxis machinery. a** Side-views of the membrane-associated chemotaxis apparatus illustrate the location of the receptor layer (Layer 1) and CheA:CheW baseplate (Layer 2). These layers are spaced ~90 Å from one another. **b** Sub-volume averaging of three *Td* strains reveals the universally conserved receptor trimer-of-dimer arrangement with 12 nm spacing between opposing trimer-of-dimer modules. Notably, density is apparent in the center of the receptor hexagons (blue arrows) and between some receptor trimer-of-dimer modules (green arrows). In general, the densities in Layer 1 of the wild-type (WT) strain are better resolved. **c** Sub-volume averages at Layer 2 reveal the organization of CheA. Density corresponding to CheA is only apparent in the two *Td* deletion mutants and CheA is arranged in a linear fashion. In this arrangement, the density between the trimer-of-dimer modules (green arrows) corresponds to the CheA P3 domain.

A *Td* strain lacking the CheR$_{like}$ domain (ΔCheR$_{like}$) reveals a significant decrease in the prevalence and size of the arrays (Fig. S4a, b). Due to the small size of the arrays in ΔCheR$_{like}$, only 194 particles were available for sub-tomogram averaging. The resulting averages are of significantly lower resolution, do not reveal the cell axis, and the CheA domains are not apparent (EMD-11386; Figs. 4E and S4b). Chemotaxis assays with ΔCheR$_{like}$ toward the attractants glucose and hemin do not show a significant change in chemotaxis behavior compared to WT (Fig. S6)[13]. However, this result is expected, as previous studies in *E. coli* demonstrate that extended arrays are only necessary for imparting cooperativity and disassembled arrays that are otherwise functional support chemotaxis[21]. These results demonstrate that the CheW-CheR$_{like}$ protein binds strongly to CheA and that the CheR$_{like}$ domain is essential for maintaining the structural integrity of assembled arrays. Given that the short linker between the CheW and CheR$_{like}$ domains necessarily places the CheR$_{like}$ domains adjacent to the ring components, it may contribute to the extra "middle density" at the ring center.

**Protein interfaces in the CheA:CheW:CheW-CheR$_{like}$ rings**. The cryo-ET experiments demonstrate that CheA is incorporated into the CheA:CheW rings in a strict linear arrangement (i.e., that CheA P5 can only occupy two of the six positions; Fig. 3c). Such an arrangement could be facilitated provided that there are three components in the ring that generate three unique interfaces. Bioinformatics analyses demonstrate that all functional Spirochaetota F2 chemotaxis systems possess a CheW-CheR$_{like}$ homolog and at least one classical CheW protein (Figs. S1 and S3). To explore the potential binding interfaces within the *Td* rings, we analyzed homology models of the classical CheW, the CheW domain of CheW-CheR$_{like}$, and the CheA P5 domain using available structures (PDB ID: 2QDL and 6S1K; Figs. 5 and S7)[11,22]. Three of the four regions with lowest sequence conservation among the three domains are located at interfaces 1–3 (Fig. 5). Alignment of the *Td* CheW and CheA P5 models to a crystal structure of *Tm* CheW in complex with *Tm* CheA P5 (PDB ID: 3UR1) further illustrates that these regions are located

at the CheW:P5 ring interfaces (Fig. 5)[23]. Mapping the variable regions onto the sequence logos of the F2 CheW domains demonstrates that they evolved different sequence patterns in these regions, with the exception of the variable region that is not located at the interaction interface (region 2, Figs. 5 and S2b). These analyses suggest that a preferred arrangement of these three domains accounts for the strict linear ordering of CheA.

**CheA arrangement and array curvature in *T. denticola***. Sub-tomogram averaging reveals that *Td* CheA forms a linear arrangement across the chemotaxis array, linking the CheA:CheW rings into extended "strands" that are held together by receptor:CheA/W interactions (Fig. 6a, b). The array densities at Layer 1 (receptors, P3, and CheR$_{like}$) produce apparent lines in the cryo-ET reconstructions that run relatively parallel to the cell axis (Fig. 6c, d)[24]. Indeed, the angle between the cell axis and these lines in the cryo-ET reconstructions is 10.4 ± 8.6°, (*n* = 26 cells), and no significant difference was found among the three *Td* strains measured (Table S1a). This arrangement allows interactions that hold the CheA:CheW strands together to occur with minimal bending (Fig. 6e, f), keeping the CheA:CheW strands relatively perpendicular to the cell axis (Fig. S8a).

*Td* cells demonstrate a significantly higher curvature of the cell membrane than other organisms investigated for the arrangement of their chemotaxis arrays thus far[4,8,25]. As a measure of comparison, the *Vc* mini-cells used in a previous study have an inner membrane curvature of 9.15 ± 4.5 μm$^{-1}$ (radius 1092 Å, *n* = 6 cells), and the *Td* cells have an inner membrane curvature of 35.8 ± 6.6 μm$^{-1}$ (radius: 279 Å, *n* = 10 cells; Fig. S8b, c and Table s1b, c). Additionally, the measured curvature of the *Td* CheA:CheW baseplate is 65.6 ± 19 μm$^{-1}$ (radius: 152 Å, *n* = 10 cells; Fig. S8b and Table S1c).

To determine the extent of bending that occurs in CheA:CheW rings that run perpendicular to the cell axis (a radius of 152 Å), we examined the CheA:CheW rings present in a crystal structure (PDB ID: 3UR1). A single ring is flat with a diameter of ~95 Å. The length across two rings connected by a dimeric CheA is 224 Å (Fig. S9a, b)[23]. The 95 Å and 224 Å rings were modeled as a

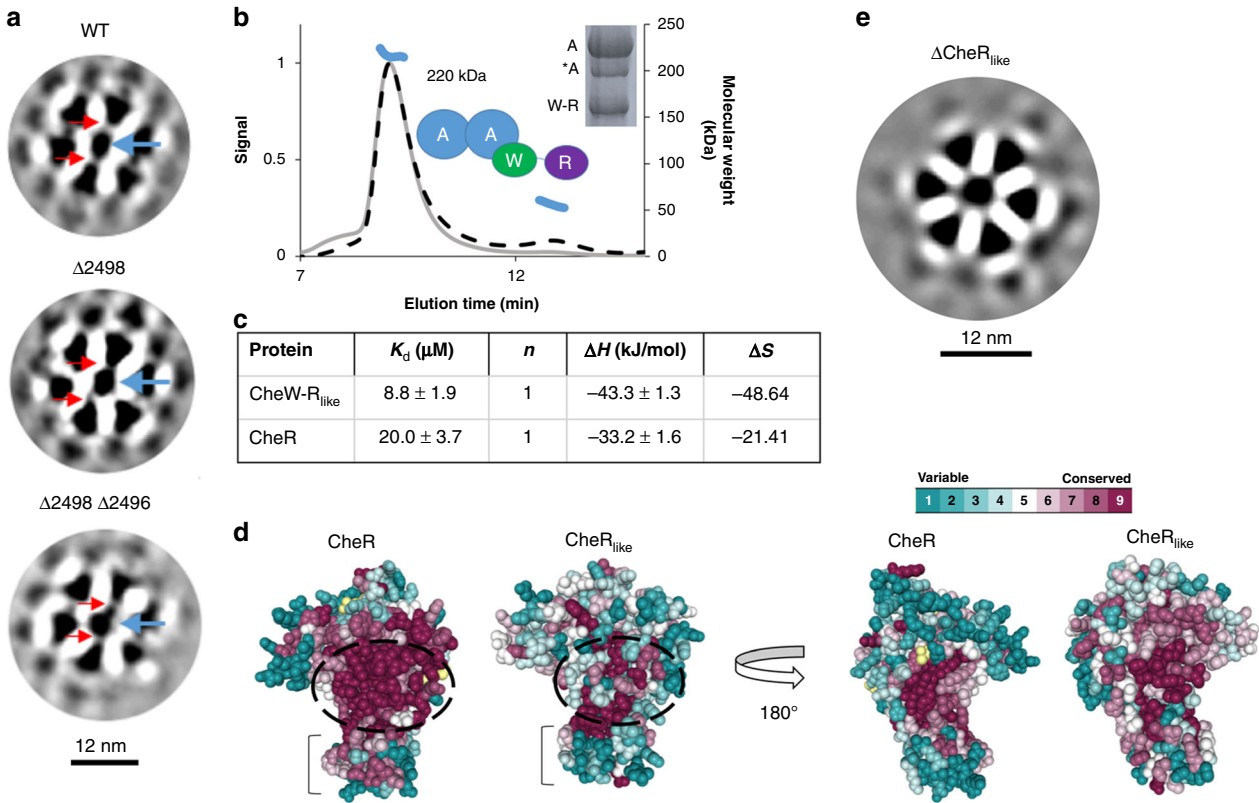

**Fig. 4 Investigations into the CheW-CheR$_{like}$ protein. a** In Layer 1, the middle density (blue arrow) extends to two positions in the CheA:CheW rings for all three *Td* strains (red arrows). **b** SEC-MALS experiments with CheW-R$_{like}$ and CheA demonstrate the presence of a ~220-kDa complex in solution, which is composed of one CheA dimer and one CheW-R subunit. A small amount of unassociated CheW-R$_{like}$ is present. Inset: SDS-PAGE analysis of the elution peak demonstrates the presence of both proteins. A small amount of degraded CheA is also present [*A]. **c** Thermodynamic binding parameters derived from the microcalorimetric titration of SAM into CheR and CheW-CheR$_{like}$. Values are reported with 90% confidence. **d** Homology models of *Td* classical CheR and the CheR$_{like}$ domain from CheW-CheR$_{like}$ were generated and the sequence conservation among F2 homologs were mapped onto the two models. The SAM binding pockets are located in black dashed circles. The sub-domain that binds receptors is indicated with a bracket. Rotating the models by 180° reveals a surface area more highly conserved in CheR$_{like}$. **e** Deletion of a CheR-like domain that is covalently attached to the C-terminus of a CheW domain generates sub-tomogram averages with diminished resolution. Source data are provided as a Source Data file.

chord in a circle with radius 152 Å. Using the equation $h = r - \sqrt{r^2 - L^2}$ (where $h$ is the height of the circular segment, $r$ is the circle radius, and $L$ is half the chord length (95 Å /2 and 224 Å /2)), the height of the circular segment is 7.6 Å and 49.2 Å, respectively (Fig. S9a, b). Therefore, a single ring and the center of two connected rings (the P3 domain) must bend by an average of 7.6 Å and 49.2 Å toward the cell membrane to accommodate the measured baseplate curvature, respectively.

**Spirochetes possess an atypical dimerization domain.** The cryo-ET results reveal density corresponding to the P3 domain, which has not been previously reported in in vivo arrays. Sequence alignments of *Td* CheA with CheA homologs from a variety of model bacteria with previously characterized chemotaxis systems indicate that, in *Td* CheA, an additional ~50 residues join the canonical dimerization domain (P3) helices (Fig. S12a)[7]. CheA homologs from other spirochete genera, including *Borrelia* and *Brachyspira*, also possess additional residues in this region (Fig. S10b)[26]. Analysis of non-redundant P3 domains from all CheA classes reveal general sequence conservation in the canonical helices but highly divergent sequences at these additional residues (Fig. S11a). Furthermore, CheA-F2 proteins contain the most residues in this non-conserved region (Fig. S11b, c). The x-ray crystal structure of the isolated *Td* P3 domain (PDB ID: 6Y1Y, Fig. 7, and Table 1) reveals that the additional residues adopt the coil-coiled motif of the classic dimerization domain with the

exception of a break in one of the helices, producing a discontinuous coiled-coil (Figs. 7 and S12a). Interestingly, aromatic residues (Phe, Tyr) cluster near the helix breakages, and unusual core packing of these residues allows for maintenance of a coiled-coil register despite a disruption of helical heptad repetition in the C-terminal helix (Fig. S12a). The net result is a distortion in the alignment of the hairpin tip, the consequence of which is currently unknown. Different orientations of Tyr83 in the two subunits produce asymmetry in the added tip region (Fig. S12b, c). Fitting the new P3 domain into an all-atom chemotaxis array that was generated for previous molecular dynamics simulations (PDB ID: 3JA6) shows that these additional helices are within ~15 Å from receptors (Fig. S12d)[12]. Additionally, the handedness of the helix connection in the *Td* P3 domain differs from that of *Thermotoga maritima* CheA and instead matches the helix connectivity of sensor kinase DHP domains[27,28].

## Discussion

Here, we reveal the protein arrangement of F2 chemotaxis arrays through cryo-ET of intact *T. denticola* (*Td*) cells. *Td* cells have the smallest average diameter (0.1–0.4 µm) of all bacteria whose chemotaxis architectures have thus far been determined[4,5,7]. The cells are thin and cylindrical, producing a cell that is polarized in shape and membrane curvature; the membrane has extremely high curvature perpendicular to the cell axis and lower curvature parallel to the axis. In accordance with this feature, the *Td* arrays

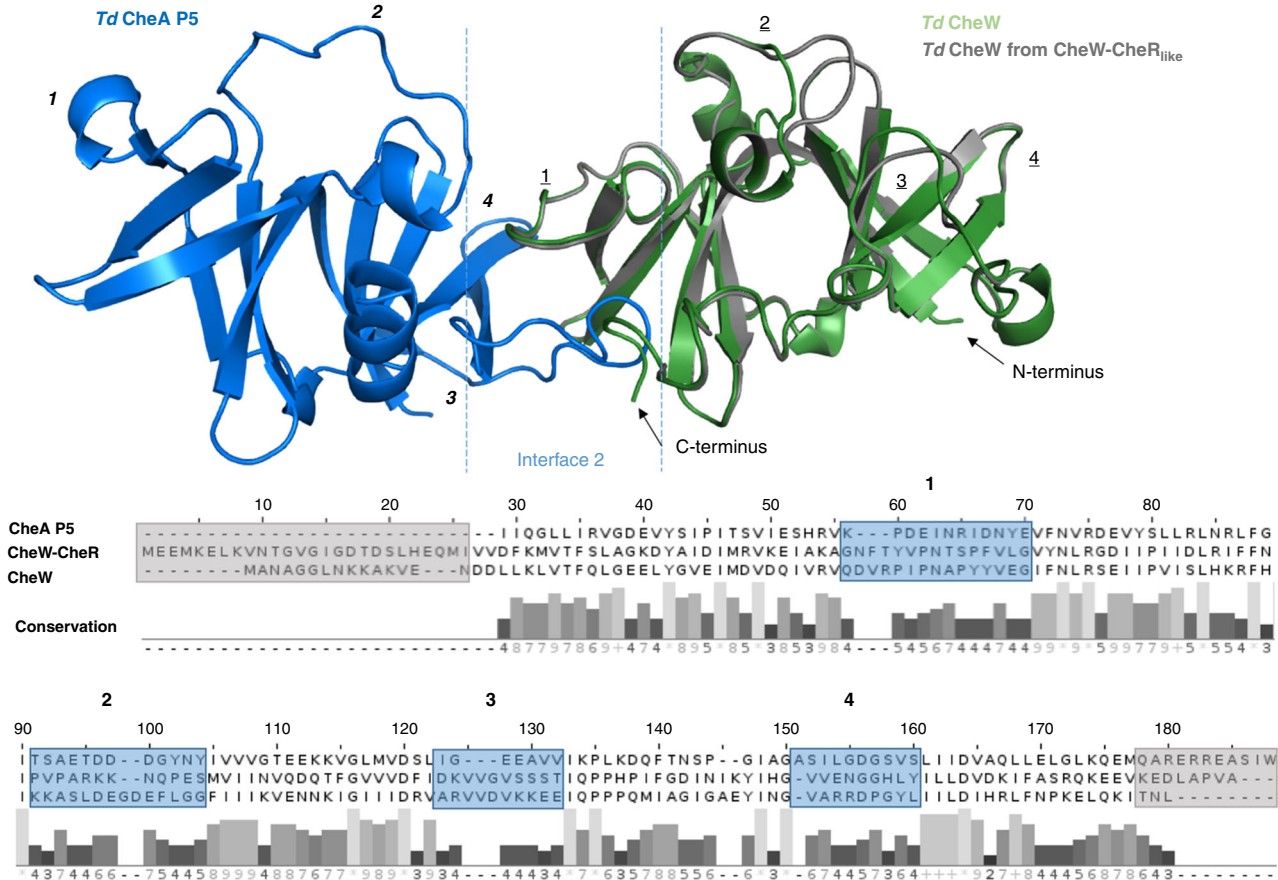

**Fig. 5 Homology models of *Td* CheA P5, CheW, and the CheW domain of CheW-CheR_{like} mapped onto a previously determined crystal structure of Thermotoga maritima CheA P5 and CheW (PDB ID: 3UR1).** Variable regions between the three domains (blue boxes) were determined by sequence alignments of the three domains followed by conservation analysis. These variable regions are located at the rings interface regions. The CheW domains also have variable N-terminal and C-terminal regions that are not complimented in P5 (gray boxes). Variable regions for P5 and the CheW domains are denoted on the homology models by italicized numbers and underlined numbers, respectively.

are polarized and assume a preferred orientation with respect to the cell axis. In this system, three proteins comprise the rings at the receptor tips: CheA, CheW, and a CheW-CheR_{like} protein. Like the *Ec* system, proteins in *Td* are integrated into the array with strict organization[8,25]. However, a linear arrangement of CheA is present that generates "strands" of rings interlinked by the CheA dimerization domain (P3). The strands run perpendicular to the axis of the *Td* cells, resulting in substantial bending of interlinked rings. Presumably, the unique binding interfaces in the CheA:CheW rings generate bent interfaces with the appropriate curvature. Furthermore, this directional distinction allows for some array contacts to follow the cell axis and remain undistorted. The deleterious effect on array integrity with the loss of the CheR_{like} domain strongly suggests that CheR_{like} plays a key role in array assembly and stabilization, perhaps through dimerization across the ring that is encouraged through close proximity in vivo. The strict linear arrangement of CheA could be facilitated by the composition of the *Td* rings; three unique protein interfaces are present in the rings and restrict CheA P5 integration (i.e., CheA P5 can only occupy these two positions in the six-member ring). Furthermore, the *Td* CheA P3 domain is clearly discernible in the sub-tomogram averages, which has not been previously observed in vivo[4,5,25]. As the CheA:CheW rings have to undergo substantial bending to accommodate the baseplate curvature, the elongated P3 domain may have evolved to stabilize CheA dimerization by increasing the interface area. It may also encourage interactions with neighboring receptors[12]. Because of the high membrane

curvature, the receptor trimer-of-dimer modules are expected to be further splayed, and the elongated P3 may compensate for the increased distance between receptors and P3.

In summary, there are several molecular features of the *Td* arrays that are not shared by the canonical system. First, *Td* has evolved three components in the CheA:CheW rings that may ensure that CheA is arranged in a strictly linear formation. This feature produces chemotaxis arrays that have a bilateral symmetry (P2), as opposed to the canonical radial symmetry (P6), and have a preferred orientation with respect to the cell axis and highly curved membrane. Second, the P2 symmetry and preferred orientation allows specific protein interaction sites to follow the path of least curvature in the cell, presumably to maintain contacts with reduced strain. Third, the arrays in *Td* include a new structural component, the CheR_{like} domain of CheW-CheR_{like}, that is crucial for maintaining array integrity. The high membrane curvature may impose substantial strain to the arrays, and the CheR_{like} domain may be needed for additional stability. Lastly, the *Td* CheA protein possesses an extended P3 domain that may increase stability of the dimerization domain that would undergo additional strain from aligning to the substantially curved CheA:CheW baseplate, and/or interact with receptors that are further splayed due to the highly curved membrane. Collectively, these features support the conclusion that they have evolved to support array formation in a highly curved membrane.

Bioinformatics analyses indicate that the unique protein features seen in *Td* are exclusive to all Spirochaetia (F2) systems.

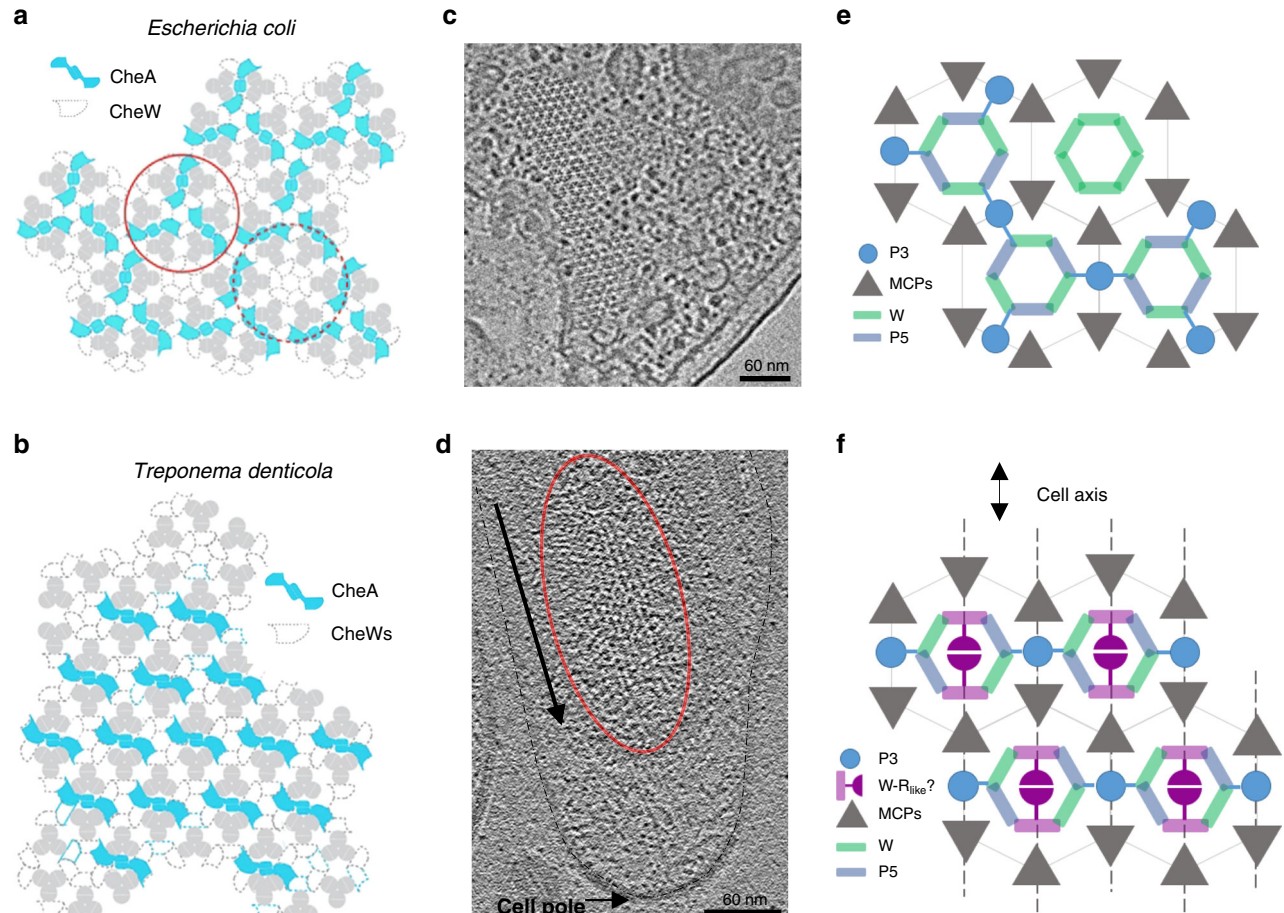

**Fig. 6 The organization of CheA in Td differs from all previously reported arrangements. a** The hexagonal arrangement of CheA in *E. coli* and other canonical chemotaxis systems. **b** In *Td*, CheA is arranged with a strict linear organization, producing "strands" of CheA:CheW rings linked by the CheA P3 domain. **c** The receptor densities in *Ec* (Layer 1) are apparent and hexagonal. Figure adapted from Briegel A. et al. (2013).[24] and reprinted with permission. **d** The densities at Layer 1 in *Td* (red oval) produce apparent lines of recetors, P3 and the middle density (CheR-like) that follow the axis of *Td* cells (black arrow). **e** *Ec* arrays possess radial P6 symmetry where all interaction interfaces are equally curved. **f** The *Td* array has a two-fold symmetry arrangement that allows receptor:CheA/W interactions that hold the CheA:CheW strands together to occur with minimal bending. Dashed lines represent the apparent lines in the tomograms. The CheR$_{like}$ domain may reside in the center of the CheA:CheW rings.

Indeed, gene deletion studies of *Bb* have shown that two CheW proteins, a classical CheW (CheW1) and a CheW-CheR$_{like}$ (CheW3), are essential for array formation and chemotactic behavior, and they possess unique regions at the ring interfaces[9]. *Bb* also possesses two CheA homologs (CheA1-F8 and CheA2-F2), but only one of the homologs (CheA2) contains an elongated P3 domain and is essential for chemotaxis and pathogenicity[26]. Therefore, we predict that a similar chemotaxis arrangement is present in *Bb*, which has a similar diameter as *Td*. However, cryo-ET experiments of *Bb* fail to produce top-view images of arrays sufficient for sub-tomogram averaging, for reasons that are unclear[7,29]. Furthermore, cryo-ET experiments of cell poles in other spirochetes have been conducted (*Treponema pallidum*[30] and *Leptospira*[31]), but top-views of arrays in these species have not been reported yet.

Unexpectedly, the placement of CheA in WT *Td* arrays could not be discerned (with the exception of the P3 domain) but was clearly visible in two *Td* mutants (Δ2498, Δ2498Δ2496). As the density corresponding to the P3 domain in the WT strain is clearly discernible, the sparse density corresponding to all other CheA domains (P1, P2, P4, and P5) is not attributed to low CheA incorporation in these arrays. These results indicate that the kinase is highly mobile or more disordered in the WT strain. In

contrast, it is more constrained when ODP (TDE2498) is deleted, suggesting that ODP directly affects array structure. However, densities in the three strains do not designate an obvious position for ODP, indicating that ODP may not be an integral component of the array. Rather, it may peripherally interact with the chemotaxis machinery or influence array architecture through other means, perhaps related to its signaling properties.

In summary, we illustrate an arrangement of the chemotaxis array that has evolved to complement the high membrane curvature and asymmetry of spirochetes. Therefore, it is likely that the behavior and characteristics of chemoreceptor arrays in general can be influenced by perturbing the shape of the cell membrane. Recent studies with *Ec* ultra-minicells (the smallest mini-cells available to date) produce densities corresponding to portions of chemoreceptors that have not been observed before[10]. However, it is possible that increased membrane curvature limits movement of the transmembrane receptors, resulting in increased receptor localization and resolution. Furthermore, the use of lipid-templating has been extensively used to assemble arrays in vitro for cryo-ET; this method reconstitutes the chemotaxis apparatus in a perfectly flat formation[11,12]. While cryo-ET experiments of in vivo *Ec* arrays consistently reveal the CheA P3 density to be too sparse to determine its position, cryo-ET of

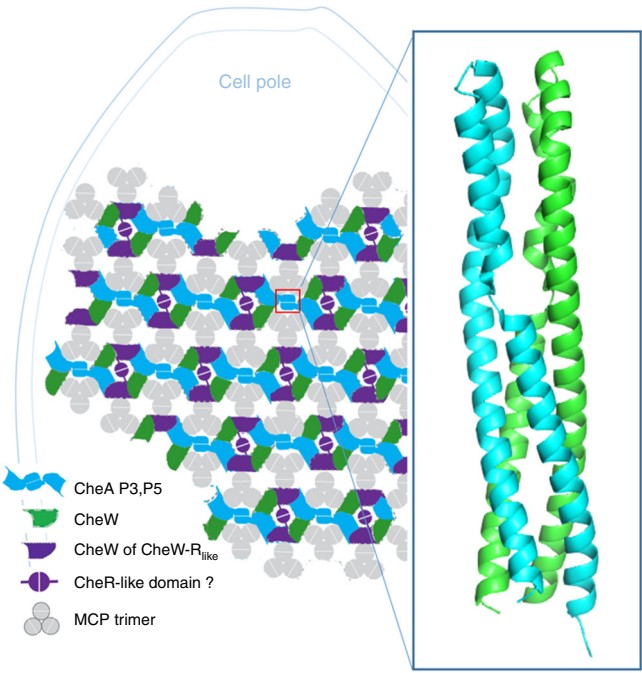

**Fig. 7 The arrangement of chemotaxis proteins in *Td*.** The linear organization of CheA produces "strands" of CheA:CheW rings that are perpendicular to the cell axis and are linked by an atypical P3 domain. While the placement of CheA is discernible in the sub-tomogram averages, the positioning of the two CheW domains in the rings is unclear but is illustrated in this model for simplicity.

Legend:
- CheA P3,P5
- CheW
- CheW of CheW-R$_{like}$
- CheR-like domain ?
- MCP trimer
- Cell pole

**Table 1 Data collection and refinement statistics for the *Td* CheA P3 domain.**

|  | (PDB ID: 6Y1Y) |
| --- | --- |
| Wavelength | 0.979100 |
| Resolution range | 75.36–1.50 (1.55–1.50) |
| Space group | C 1 2 1 |
| Unit cell | a, b, c: 101.64, 28.77, 82.90 α, β, γ: 90, 114.75, 90 |
| Total reflections | 53,542 |
| Unique reflections | 35,178 (3490) |
| Multiplicity | 6.9 (6.2) |
| Completeness (%) | 96.72 (99.31) |
| Mean I/sigma(I) | 14.7 (5.4) |
| R-merge | 0.063 (0.290) |
| R-meas | 0.075 (0.350) |
| R-pim | 0.040 (0.193) |
| CC1/2 | 0.99 (0.94) |
| Reflections for refinement | 33,864 (3361) |
| Reflections for R-free | 1308 (143) |
| R-work | 0.187 (0.205) |
| R-free | 0.211 (0.244) |
| No. non-H atoms | 2344 |
| Macromolecules | 2028 |
| Ligands | 0 |
| Solvent | 316 |
| Protein residues | 253 |
| RMS(bonds) | 0.006 |
| RMS(angles) | 0.93 |
| Ramachandran favored (%) | 100 |
| Ramachandran allowed (%) | 0.00 |
| Ramachandran outliers (%) | 0.00 |
| Rotamer outliers (%) | 0.40 |
| Clashscore | 6.87 |
| Average B-factor | 19.1 |
| Macromolecules | 17.2 |
| Solvent | 28.2 |

in vitro lipid-templated *Ec* arrays clearly defines the P3 position. If the P3 domain does engage receptors, this discrepancy may suggest that lipid-templating abrogates this native behavior. Furthermore, it is unclear how the *Td* chemotaxis architecture affects cooperative behavior in the arrays; linear CheA:CheW strands are connected only through receptor interactions. Investigations into this system may reveal alternative mechanisms for cooperative behavior than those that have been reported. Collectively, our results illustrate the importance of investigating transmembrane systems in situ and show that examining systems in non-model organisms can lead to unexpected advances to our understanding of the remarkable signaling systems of bacterial chemotaxis.

## Methods

**Bacterial strains, culture conditions, and oligonucleotide primers**. *Treponema denticola* (*Td*) ATCC 35405 (wild-type) was used in this study. The *Td* deletion mutants, Δ2498 and Δ2498Δ2986, were generated in a previous study[13]. Cells were grown in tryptone-yeast extract-gelatin-volatile fatty acids-serum (TYGVS) medium at 37 °C in an anaerobic chamber in presence of 85% nitrogen, 10% carbon dioxide, and 5% hydrogen[32]. *Td* mutants were grown with an appropriate antibiotic for selective pressure as needed: erythromycin (50 μg/ml) and gentamycin (20 μg/ml). *Escherichia coli* 5α strain (New England Biolabs, Ipswich, MA) was used for DNA cloning. The *E. coli* strains were cultivated in lysogeny broth (LB) supplemented with appropriate concentrations of antibiotics. The oligonucleotide primers for PCR amplifications used in this study are listed in Table S3. These primers were synthesized by IDT (Integrated DNA Technologies, Coralville, IA).

**Construction of a CheR truncated mutant (ΔCheR$_{like}$)**. *TDE1492::ermB* (Fig. S13) was constructed to replace the CheR-like domain (8781–1308 nt) in *TDE1492* with a previously documented erythromycin B resistant cassette (*ermB*)[33]. The *TDE1492::ermB* vector was constructed by two-step PCR and DNA cloning. To construct this vector, the 5′ end of *TDE1492* region and the downstream flanking region were PCR amplified with primers P$_1$/P$_2$ and P$_3$/P$_4$, respectively, and then fused together with primers P$_1$/P$_4$, generating Fragment 1. The Fragment 1 was cloned into the pMD19 T-vector (Takara Bio USA, Inc, Mountain View, CA). The *ermB* cassette was PCR amplified with primers P$_5$/P$_6$, generating Fragment 2. The Fragment 2 was cloned into the pGEM-T easy vector

(Promega, Madison, WI). The Fragment 1 and 2 were digested using *Not*I and ligated, generating the *TDE1492::ermB* plasmid. The primers used here are listed in Table S3. To delete *TDE1492*, the plasmid of *TDE1492::ermB* was transformed into *Td* wild-type competent cells via heat shock for 1 min at 50 °C[34]. Erythromycin-resistance colonies that appeared on the plates were screened by PCR for the presence of *ermB* and absence of *TDE1492 (781–1308 nt)* gene. The PCR results showed that the *TDE1492 (781–1308 nt)* gene was replaced by *ermB* cassette as expected (Fig. S13). One positive clone (*ΔTDE1492*) was selected for further study.

**Bioinformatics software and resources**. The datasets used in the bioinformatics analysis were built using data from Microbial Signal Transduction Database v3 (MiST3) accessed February 2020[15] and the Genome Taxonomy Database v89 (GTDB)[16]. We built custom scripts using TypeScript-3.7.5 and NodeJS-12.13. To make the scripts, we also used packages publicly available at the node package manager repository (npm): we used RegArch-1.0.1[35] to separate CheW-CheR$_{like}$ from other CheWs, gtdb-local-0.0.12 (https://npmjs.com/package/gtdb-local) to use the GTDB taxonomy, Phylogician-TS-0.10.1-4 (https://npmjs.com/package/phylogician-ts) to visualize and manipulate the phylogenetic trees, BioSeq-TS-0.2.4 (https://npmjs.com/package/bioseq-ts) to handle protein sequences and MiST3-TS-0.7.6 (https://npmjs.com/package/mist3-ts) to access MiST3 API. Multiple sequence alignments were produced using L-INS-I algorithm from the MAFFT package. To reduce redundancy in sequence datasets we used CD-HIT v4.6[36] with unaligned sequences and Jalview[37] with aligned sequences. RAxML v8.2.10[38] was used to perform phylogenetic reconstructions, and low support branches in the phylogenetic trees were collapsed with TreeCollapseCL4[39]. Sequence logos were built using Weblogo 3.7[40].

**Bioinformatics scripts and pipelines**. We collected all information on the proteins classified as CheA (96,434) and CheW (134,165). To process this dataset we built several scripts and pipelines to produce the tables, figures, and datasets used in this analysis (Fig. S14). See Code Availability for access to the custom scripts.

**Chemosensory profile of Spirochaetotas**. The "spiro-pipeline" selects all genomes from Spirochaetota phylum using gtdb-local package to access GTDB v89,

then it filters only the genomes that are also present in MiST3 database. It collects the information on MiST for each genome and appends the complete taxonomy information from GTDB and signal transduction profiles. Finally, the pipeline builds the table with the information in markdown (Dataset 2).

**Chemosensory profile of genomes with at least one CheA-F2.** The "chea-pipeline" starts from the raw dataset taken from MiST3 database with information on 96,434 CheA genes. Based on MiST3 classification it selects the genomes with at least one CheA-F2 sequences and fetches information about these genomes. At this step, it also checks with the list generated by the "wr-pipeline" of the genomes containing CheW-CheR$_{like}$. It proceeds to append chemosensory information for each genome and GTDB taxonomy. At this step the pipeline splits into four pathways, where one parses the information to build the Dataset 1 and the other three build FASTA formatted files with sequences from: CheA-F2, CheR-F2, CheW-CheR$_{like}$ and all CheWs belonging to genomes with at least 1 CheA-F2. We noticed that MiST3 currently misclassifies some CheR proteins, so we used RegArch to filter out false positives. We also used RegArch to separate CheW and CheW-CheR$_{like}$ sequences as MiST3 classifies both as CheW (scaffold). The RegArch definitions can be found in the script "regArchDefinitions.ts" of the source code. See Code Availability for access to the source code.

We selected genomes from the MiST3[15] databse that contain at least 1 CheA-F2 (Dataset 1) and we found no genomes with more than 1 CheA-F2 per genomes. Thus, there are 306 F2 systems in the MiST3 databse. All 306 genomes are Spirochaetota with three exceptions: two of them belong to Acidobacteriota phylum, and one from Planctomycetota, which suggest that the presence of an F2 system in these genomes (outside of the Spirochaetota phylum) is the consequence of lateral gene transfer. Of the 306 genomes with at least one CheA-F2, only the following lack the CheW-CheR$_{like}$ protein: the three non-Spirochaetota genomes mentioned previously, 40 genomes of the Brachyspirae class, and three genomes from the order Borreliales (Fig. S1). Interestingly, the Borreliales genomes do appear to have the CheW-CheR$_{like}$ gene, except there is no gene product associated with them in the MiST3 database.

**Classification of CheW.** CheW proteins are not classified in MiST3. In order to make comparisons between the sequences of CheW domains in F2 systems, we must first select only CheW-F2. We first assign to the F2 class all the canonical CheW found in genomes with a single CheA-F2. Contrary to the F2 systems where the canonical CheW is not present in the chemosensory gene clusters, other classes do contain their canonical CheW within the rest of the gene cluster. CheW found within 5 genes from a classified CheA were assigned to the same class as CheA. To perform this classification, we selected the 598 full-length CheW sequences generated by the chea-pipeline. We used CD-HIT to remove 405 redundant sequences (-c 1). Next, we ran the "classify-w" pipeline on the remaining sequences (193). The pipeline reads the identifiers of the sequences and fetches the chemotaxis profile from MiST3 for each genome. It classifies CheWs as F2 classes if there is only 1 CheA of the class F2 in the profile. Next, it fetches the gene neighborhood (5 genes up and downstream) of the remaining CheWs and assigns a matching class if there is a classified CheA within these genes. We also aligned the sequences (193) with the L-INS-I algorithm of the MAFFT package, produced a phylogeny with 1000 rapid bootstrap using RAxML (-f a -m PROTGAMMAIAUTO -N 1000) and collapsed the phylogeny using TreeCollapse4 at 50% bootstrap. We mapped the CheW classification to the CheW tree in Fig. S3. We expanded the F2 classification to the 74 sequences within the branch with only CheW-F2 sequences.

**Comparison of the CheW domains in CheA-F2, CheW-CheR$_{like}$' and CheW-F2.** We put together the sequences from CheA-F2 and CheW-CheR$_{like}$ (both trimmed by the PFAM model for the CheW domain, already annotated in MiST3) and the full sequence of the CheW-F2 selected in the previous step. We then use L-INS-I algorithm to align the sequences and Jalview to manually inspect and eliminate identical redundant sequences. Finally, we trimmed the whole alignment based on the boundaries of CheA-F2 and CheW-CheR$_{like}$ and eliminated one incomplete sequence: GCF_000413015.1-HMPREF1221_RS07250. The final alignment had a total of 206 sequences: 73 CheA, 59 CheW-CheR$_{like}$, and 74 CheW. We separated the alignments into individual files and built sequence logos to summarize the amino-acid diversity in each position for each group (Fig. S2b).

**Comparison of CheR domains.** First, we added together the trimmed part matching the CheR domain of the CheW-CheR$_{like}$ sequences and the 292 sequences of the CheR protein. We then aligned the 550 sequences using L-INS-I. We used Jalview to inspect the alignment and remove identical redundant sequences. The final alignment had the CheR domain of 83 CheR and 83 CheW-CheR$_{like}$ proteins. We separated the alignments into individual files and generated independent sequence logos (Fig. S2a).

**Analyses of CheA P3 domains.** The "p3-pipeline" pipeline processes the data for the analysis of the length of P3 domains of all CheA homologs in the MiST3 database. It reads the information for all 96,434 CheAs in the MiST3 database, trims the sequence matching the Pfam mode H-kinase_dim and builds FASTA formatted datasets for each chemotaxis class. For each dataset, we used CD-HIT

with 75% identity cut-off and aligned them using the L-INS-I algorithm from MAFFT. Using Jalview we manually inspect the alignment and edited the alignment to remove divergent sequences that opens major gaps in the alignment. We removed 6 F1 sequences, 37 F7 sequences, 20 F8 sequences, and 5 F9 sequences. Then we merged the alignment using mafft-profile with each dataset as a seed alignment in a single shot (Fig. S11a). We selected the non-conserved central region of the alignment and measured the number of amino-acids in each sequence (Fig. S11b).

**Cryo-ET and sub-tomogram averaging of *T. denticola* chemotaxis arrays.** Cells were concentrated by centrifugation, and a 1/10 dilution of protein A-treated 10-nm colloidal gold solution (Cell Microscopy Core, Utrecht University, Utrecht, The Netherlands) was added to the cells and mixed by pipeting. In all, 3 μL aliquots of the cell suspension were applied to glow-discharged R2/2 200 mesh copper Quanti-foil grids (Quantifoil Micro Tools, GmbH), the sample was pre-blotted for 30 s, and then blotted for 2 s. Grids were pre-blotted and blotted at 20 °C and at 95% humidity. The grids were plunge-frozen in liquid ethane using an automated Leica EM GP system (Leica Microsystems).

Data collection was achieved on a Titan Krios transmission electron microscope (Thermo Fisher Scientific) operating at 300 kV. Images for three strains (WT, Δ2498, Δ2498Δ2986) were recorded with a Gatan K2 Summit direct electron detector with a GIF Quantum energy filter (Gatan) operating with a slit width of 20 eV. Images were taken at a magnification of ×42,000, which corresponds to a pixel size of 3.513 Å. Tilt series were collected using SerialEM with a modified bidirectional tilt scheme (−20° to 60°, followed by −22° to −60°) with a 2° increment. Images for the ΔCheR$_{like}$ strain were recorded with a Gatan K3 Summit direct electron detector equipped with a GIF Quantum energy filter (Gatan) operating with a slit width of 20 eV. Images were taken at a magnification of ×26,000, which corresponds to a pixel size of 3.27 Å. Tilt series were collected using SerialEM with a bidirectional dose-symmetric tilt scheme (−60° to 60°, starting from 0°) with a 2° increment. For all strains, the defocus was set to −6 μm and the cumulative exposure per cell was 100 e-/A$^2$.

Bead tracking-based tilt series alignment and drift correcting were done using IMOD[41]. CTFplotter was used for contrast transfer function determination and correction[42]. Tomograms were reconstructed using simultaneous iterative reconstruction with iteration number set to 6 and binning set to 6. The resulting pixel size of the tomograms was 7.026 Å (for WT, Δ2498, and Δ2498Δ2986 strains) and 6.54 Å (for the ΔCheR$_{like}$ strain). Dynamo was used for manual particle picking and sub-tomogram averaging[43,44]. As only top- and bottom- views of the arrays could be used for sub-tomogram averaging, the resulting maps are anisotropic in resolution (Fig. S15a–d). Half maps were generated by the "gold standard procedure" on Dynamo. FSC calculations and local resolution filtering were done on Relion (Fig. S16a–d). The reported resolution was calculated at FSC = 0.3 within a masked region that contains the center receptor hexagon and one CheA:CheW ring (Fig. S4a). The size of the mask was 30 px × 30 px × 40 px with a soft edge of 10 px.

**Chemotaxis assays.** The chemotaxis of *T. denticola* was tested by capillary assay[13]. Log-phase cultures of *T. denticola* were centrifuged at 5000×g for 7 min and supernatants were discarded. Cell pellets were resuspended in motility buffer (0.15 M NaCl, 10 mM NaH2PO4, 0.05 mM EDTA, 1% BSA, and 0.5% methylcellulose). The motility buffer was equilibrated in anaerobic chamber overnight. The final bacterial cell concentration was adjusted to 1 × 10$^9$ cells/ml. Capillary tubes (0.025 mm inner diameter) were filled with either 0.5 mM hemin or 0.5 mM glucose in the motility buffer and sealed with vacuum silicone grease (Dow Corning, cat# Z273554-1EA) before they were inserted into bacterial suspensions (500 μl each). After incubation in anaerobic chamber at 37 °C for 2 h, the contents of each capillary tube were transferred to a new microcentrifuge tube and cell numbers were enumerated using a Petroff-Hausser counting chamber (Hausser Scientific, Horsham, PA). For the non-gradient control, capillary tubes were inserted into bacterial suspensions containing either 0.5 mM hemin or 0.5 mM glucose. The bacterial counts of each strain were normalized to those in the non-gradient control. For each strain, five capillary tubes were included, and three independent experiments were conducted, and the results are represented as the mean of cell numbers ± standard error of the mean (SEM).

**Purification of CheA, CheA P3, CheW-CheR$_{like}$, and TDE2496 proteins of *T. denticola*.** DNA segments encoding the CheA P3 domain, CheW-CheR$_{like}$ protein, and TDE2496 in *T. denticola* were PCR amplified from *Td* genomic DNA using a forward oligonucleotide encoding an NdeI restriction site and a reverse primer encoding an EcoRI restriction site. The CheA protein was amplified using a forward oligonucleotide encoding an NdeI restriction site and a reverse primer encoding an BamHI restriction site. The PCR products were treated with the appropriate restriction enzymes, purified, and ligated into a pet28a plasmid with a poly-Histidine tag and kanamycin resistance marker. The plasmids were transformed into *Escherichia coli* BL21-DE3 cells and 4–8 L of cell culture were grown at 37 °C until an O.D. of 0.6 was reached. The flasks were cooled to 21 °C and 1 mM of IPTG was added to the culture. The cells were harvested after 16 h of growth. The cells were lysed via sonication in lysis buffer (50 mM Tris pH 7.5, 150 mM NaCl, 5 mM Imidazole) while cooled on ice. The lysate was centrifuged at 20,000×g

for 1 h at 4 °C. The lysate was then run over a gravity-flow purification column containing 3 ml of Nickel-NTA resin. The resin was washed with 10 ml wash buffer (50 mM Tris pH 7.5, 150 mM NaCl, 20 mM Imidazole) and the protein was eluted with 10 ml elution buffer (50 mM Tris pH 7.5, 150 mM NaCl, 200 mM Imidazole) and collected in 1 ml fractions. The fractions were assessed for protein concentration via Bradford reagent and the fractions containing protein were run on a size-exclusion s75 and s200 column systems that monitored absorbance at 280 nm and collected 6 ml fractions. Fractions that contain CheA were concentrated to ~20 mg/ml via centrifugation in a protein concentrator containing a regenerated cellulose filter with a 50 kDa molecular-weight cut-off (MWCO). Fractions that contain CheA P3, CheW-CheR$_{like}$ and TDE2496 were concentrated with a 10 kDa MWCO filter to 32 mg/ml, 11 mg/ml, and 7 mg/ml, respectively. The protein solutions were aliquoted, flash frozen in liquid nitrogen, and stored at −80 °C. For CheW-CheR$_{like}$, CheA, and CheA P3 domain, the purifications were prepared at ambient temperatures. For TDE2496, the purification was prepared at 4 °C.

**Size-exclusion chromatography coupled with multi-angle light scattering.** Multi-angle light scattering (MALS) coupled with reverse-phase chromatography experiments were used to determine the molecular weights of CheA, CheW-R$_{like}$, and their complex at 25 °C. Each sample was prepared at a final protein concentration of 10 mg/mL. All samples were buffer exchanged into the column running buffer (50 mM MOPS pH 7.5, 150 mM KCl, and 5 mM MgCl$_2$) to prevent contributions of buffer components from complicating the molecular weight calculations. The CheW-R$_{like}$ protein was incubated for 15 min with 5 mM DTT to cleave any interdimer disulfide bonds prior to buffer exchanging. The mixed sample was prepared in a 1:1 molar ratio of CheA: CheW-R$_{like}$. In all, 40 μL of each sample was injected onto a GE S200 Increase (10/300) column pre-equilibrated at room temperature. A BSA standard at 5 mg/ml was used as a calibration control. GraphPad Software's Prism8 program was used for data analysis and molecular weight calculations.

**Size-exclusion chromatography followed by SDS-PAGE.** Size exclusion chromatography of co-incubated CheA and CheW-CheR$_{like}$ followed by SDS-PAGE was performed to confirm co-elution of the proteins. The proteins were incubated in buffer (50 mM MOPS pH 7.5, 150 mM KCl, 5 mM MgCl$_2$) with excess CheW-CheR$_{like}$ and 500 μl of the mixture was injected into a GE s200 Increase (10/300) column pre-equilibrated with the protein buffer. The elutant was collected in 1 ml fractions, concentrated to ~100 μl using a 50-kDa MWCO centrifuge concentrator, and 15 μl of each fraction was ran on a precast 4–20% SDS-PAGE gel.

**Native mass spectrometry.** Purified CheR and CheW-R$_{like}$ were diluted to 2 mg/ml in 40 μl buffer (50 mM Tris pH 7.5, 150 mM NaCl) and incubated in 5 mM DTT for 1 h at 25 °C. The proteins were then buffer exchanged into mass spectrometry buffer (25 mM ammonium acetate pH 7.4) immediately before analysis. In all, 100 μM S-adenosylmethionine (SAM) was added to the protein samples and incubated for 5 min at 25 °C, and the samples were analyzed in triplicates.

Native MS analyses were carried out in an Impact qTOF mass spectrometer from Bruker (Germany) equipped with a nano-electrospray source. The mass spectrometer was operated in positive ionization mode using the following parameters: capillary voltage 1400 V, drying gas temperature 120 °C, drying gas flow rate 2 L min$^{-1}$. Transfer of the ions was achieved using a quadrupole and collision cell energy of 3 and 20 eV, respectively and a pre-pulse storage and transfer time of 25 μs and 190 μs, respectively. Mass spectra were collected in profile mode using a mass range of 500 to 8000 $m/z$. MS control and data acquisition and analysis were performed using QTOF control and data analysis software (Bruker Daltonics). Molecular mass determinations were performed using the "Maximum Entropy" algorithm of the DataAnalysis software.

**Isothermal calorimetry.** Measurements were conducted on a TA Instruments Affinity ITC Low Volume calorimeter at 25 °C. The proteins were first incubated in 5 mM DTT for 30 min at 25 °C and then exchanged into analysis buffer (50 mM Tris pH 7.5, 150 mM NaCl). S-adenosylmethionine (SAM) was also dissolved in analysis buffer to 1 mM. 200 μl of ~200 μM protein (determined by UV measurements) was placed into the sample cell and 200 μl of SAM was placed into a 250 μl syringe. The experiments were conducted by injecting 5 μl aliquots of SAM into the cell with 200 s between injections. The data were analyzed using NanoAnalyze v.3.11.0 using the Independent binding model. For all samples, heat exchanges of that occur after SAM saturation were subtracted from the titration data. Based on previous experiments[17,19] and mass spectrometry data (see section above) the stoichiometry of SAM binding to CheR and CheW-R$_{like}$ is 1:1 and hence the active fraction of protein in the cell was varied to reflect stoichiometric binding as the protein concentration was estimated by UV absorption using a theoretical absorption coefficient. The active protein concentration was calculated to be ~125 μM and ~160 μM for CheR and CheW-R$_{like}$, respectively.

**Molecular modeling of *Td* CheR and the CheR$_{like}$ domain.** Homology models of the CheR and CheR$_{like}$ domain were generated by SWISS-modeler using a crystal structure of the classical CheR protein from Salmonella typhimurium (PDB ID: 1AF7), and have a QMEAN of −2.45 and −4.64, respectively. The models and

sequences of all F2 CheR proteins and CheR$_{like}$ domains were then used in a ConSurf analysis (Consurf server) to map residue conservation onto the models.

**Quantification of cell curvature.** The cell curvature of *Td* whole cells and *V. cholera* minicells was quantified by analyzing images of cross-sections of the cells where top views of chemotaxis arrays are present. For *Td* cells, the inner membrane curvature and CheA:CheW baseplate curvature was quantified. For *Vc* minicells, the inner membrane curvature was quantified. These images were pre-processed with Fiji by placing points along the desired area with a distance of 10 nm between each point. The curvature of the inner membrane was measured with a pre-built plugin for python, called Sabl_mpl, written by Jewett, A. from the Jensen lab (Pasadena, CA)[45]. The "measure 3-point curvature" function was used to select three adjacent points along the inner membrane of the cell and calculate the radius of these points. The radius of the three selected points allowed for the calculation of the local curvature by dividing 1 with the measured radius (1/R = c). This was repeated for all points with a "sliding window" approach, where the second point of the initial three points would become the first point, until the desired area was covered.

**Quantification of array alignment to the cell axis.** The angle between the strands of CheA:CheW rings and the *Td* cell axis was quantified using Image J software. 2D images from reconstructions that clearly locate the orientation of the strands and cell axis were chosen for analysis. First, a straight line was drawn from the cell pole down the axis of the cells. Then, a second line was drawn through one of the strands in the array and the angle between the two intersecting lines was quantified. In some cases, the angle was too small (<3°) to be accurately determine so the angle was annotated as 0°.

**Residue conservation and molecular modeling of *Td* CheW, CheA P5, and the CheW domain of CheW-CheR$_{like}$.** The protein sequences of the two *Td* CheW domains and *Td* P5 were aligned (Clustal Omega), conservation was calculated based on the alignment (JalView[37]), and the highest variable regions were selected based on conservation (10+ adjacent residues with conservation scores <8). Homology models of *Td* CheA P5, CheW, and the CheW domain of CheW-CheR$_{like}$ were generated via the Swiss-Model server using complete residue sequences of each protein as a target[46]. The CheW protein from Thermoanaerobacter tengcongensis (*Tt*) (PDB ID: 2QDL) had the highest percent identity to the CheW proteins (37 and 31%) and was therefore used as the structural template. The resulting homology models for *Td* CheW and the CheW domain of CheW-CheR$_{like}$ had a QMEAN of −1.3 and −1.2, respectively. The P5 structure from *Escherichia coli* produced the best homology model for *Td* CheA P5, with QMEAN −0.98 (PDB ID: 6S1K). The homology models were then aligned to the CheW protein and P5 domain in a crystal structure containing *Thermotoga maritima* CheW and CheA P4P5 in complex using PyMol (PDB ID: 3UR1).

**Crystallization and structural determination of the P3 domain of *T. denticola* CheA.** The isolated P3 domain was concentrated to 32 mg/ml and crystallized via hanging drop in 0.1 M Imidazole pH 7.0, 25% PEG 400 using a 1:1 ratio of protein solution to crystallization solution with a final volume of 3 μl. Crystals were apparent within eight h but increased in size over 3 days. Crystals were manually picked up in loops, flash cooled and shipped in liquid nitrogen to a beamline (APS, line NE-CAT 24-ID-C, Dectris Pilatus 6M-F Pixel Array detector). The crystals diffracted to ~1.3 Å with a C2 symmetry and data was cut-off at 1.5 Å. The diffraction data was scaled and integrated using XDS[47], and phased by molecular replacement with Phaser MR using ab initio search models generated through the QUARK server and then ran through the AMPLE pipeline on the CCP4 web server[48–50]. Model improvement was done by several rounds of manual model improvement in COOT followed by automated refinement using Phenix Refine software[51,52].

**Statistics and reproducibility.** The sub-tomogram average of the *T. denticola* WT strain was generated using 11 cells and 728 sub-tomograms (Figs. 3b, 3c, 4a, S4a, b, and 8a). The sub-tomogram average of the *T. denticola* Δ2498 strain was generated using 10 cells and 894 sub-tomograms (Figs. 3b, c, 4a, and S4a, b). The sub-tomogram average of the *T. denticola* Δ2498 Δ2496 strain was generated using 10 cells with 554 sub-tomograms (Figs. 3b, c, 4a, and S4a, b). The sub-tomogram average of the *T. denticola* ΔCheR-like strain was generated using 5 cells and 194 sub-tomograms (Figs. 4e and S4a, b). Figure 3A consist of a micrograph that is representative of 31 cells. Figure 6d is a micrograph that is representative of 26 cells. Figure S8b consist of a micrograph that is representative of 26 cells. Figure S8c consist of a micrograph that is representative of 6 cells. Figure S13 is representative of three experiments.

**Reporting summary.** Further information on research design is available in the Nature Research Reporting Summary linked to this article.

## Data availability
Data supporting the findings of this manuscript are available from the corresponding author upon reasonable request. A reporting summary for this Article is available as a

Supplementary Information file. The cryo-ET sub-tomogram averages that support these findings are deposited in the Electron Microscopy Data Bank (EMDB) with accession codes EMD-11385, EMD-11381, EMD-11384, and EMD-11386. The protein x-ray crystallography structure that supports these findings is deposited in the Protein Data Bank (PDB) with accession code 6Y1Y. The sub-tomogram average pictured in Figure. 1c, d is derived from a public repository (EMD-4991)[25]. Source data are provided with this paper.

## Code availability

All custom-made scripts and instructions on how to reproduce the bioinformatics data are available on GitLab (https://gitlab.com/lab-notebook/treponema).

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

## Acknowledgements

We thank Alister Burt (Universite Grenoble Alpes), Dr. Frederic Bonnet (Netherlands Centre for Electron Nanoscopy, NeCEN), and Dr. Ludovic Renault (NeCEN) for assistance with processing of the sub-tomogram averages. We also thank Dr. Robbert Q. Kim at the Leiden University Medical Center (LUMC) Protein Facility for help with the ITC experiments, and Dr. Elena Domínguez-Vega and Prof. Manfred Wuhrer for the native MS analysis. This work is part of the research program National Roadmap for Large-Scale Research Infrastructure 2017–2018 with project number 184.034.014, which is financed in part by the Dutch Research Council (NWO). This work was funded by a grant from the National Institutes of Health: R35GM122535 awarded to B.R.C., R01AI078958 and R01DE023080 to C.L., and by the European Union under a Marie-Sklodowska-Curie COFUND LEaDing fellowship to ARM. We thank the Netherlands Centre for Electron Nanoscopy (NeCEN) for access to cryo-ET data collection facilities, and NE-CAT at the Advanced Photon Source for access to x-ray crystallography data collection facilities. NE-CAT is supported by NIH/NIGMS awards P30 GM124165 and S10 RR029205.

## Author contributions

A.R.M., D.R.O., K.K., Z.A.M., C.L., B.R.C., and A.B. designed research; A.R.M., D.R.O., K.K., and Z.A.M. performed research; A.R.M., D.R.O., W.Y., K.K., Z.A.M., and A.S. analyzed data; and A.R.M., D.R.O., and A.B. wrote the paper with input from all authors.

## Competing interests

The authors declare no competing interests.
