## [Peer Review File · Nature Communications]

REVIEWER COMMENTS

Reviewer #1 (Remarks to the Author):

Previous work, done in large part by Prof. Briegel, has characterized by cryo-EM the chemosensory arrays of a significant number of bacteria. Here the authors have studied the chemosensory array of *Treponema denticola* and found that it was different to the so far known systems. In analogy to the canonical systems there was also a hexagonal arrangement but CheAs show a linear arrangement. Bioinformatics studies show that the system belonged to category f2 and that members of this category are basically spirochaetes. The f2 gene clusters are characterized by the presence of a novel signaling protein, namely a fusion between CheW with a CheR like domain. The densities in cryo EM images corresponding to this CheW-CheR like protein have been identified. The authors argue that this protein is the reason for the existence of this unusual arrangement and the capacity of the arrays to support extreme membrane bending, which is necessary to accommodate chemosensory arrays in bacteria with a spiral topology. This is a very interesting study combining bioinformatics, biochemistry and X-ray crystallography with cryo-EM. This study will be of impact in the scientific community since it is an important contribution in the efforts to explore the diversity of chemosensory signaling. However, this referee has a number of concerns that need to be addressed.

1. Reading this manuscript one gets the impression that it contains a large amount of data that in the end dilute the core message. In this reviewer's mind the core message is that the presence of a novel protein, the CheW-CheR like fusion, that generates hexagonal arrays that are characterized by a linear CheA arrangement, which in turn permits accommodating arrays in bacteria with a spiral topology, characterized by strong bending. The initial bioinformatic part is clear and streamlined and this referee feels that attempts could be made to streamline the remaining part of the manuscript in a similar fashion; that involves removing data that do not contribute to the key story and expanding on the novel protein, using both *in vivo* and *in vitro* approaches. For example, the authors report data from mutants in Tde2498, a previously reported oxygen sensor, and Tde2496, a protein composed of solely a MCP signaling domain. The contribution of these data to the overall story is not clear; data show that removing both proteins alters array structure, indicating that both proteins are present in the array.

2. Since the CheW-CheR like protein is firstly novel and secondly central to the story, more research needs to be dedicated to this protein, both in the *in vivo* and *in vitro* level. The authors use a delta CheR like mutant, but dedicate only a few lines on the cryo-EM images on page 7. A central question is whether these arrays support bending? Maybe present an image in a similar position like that of the WT in Fig. S10? Another central question is how the chemotactic behavior of the wt compares to that of the delta CheR like mutant. Are the formed arrays in the delta CheR like protein functional? Chemotaxis assays should be conducted.

3. The weakest part of this manuscript is the biochemical part. The experimental procedures for some of the experiments are lacking, such as for example the buffer and experimental conditions used for the determination of the oligomeric state (Fig. 2D), MALS (Fig. S6B, in fact SEC-MALS was used) or the experiments that led to Fig. S6F. Some more specific points are:

Fig. S6A: Phosphatases are frequent contaminants in recombinantly purified proteins. Control experiments are to be conducted to exclude the possibility that the reduction seen is not due to the action of contaminants. These data are no clear evidence that both proteins interact. There is a wealth of more direct methods to study interaction.

6B: AU, which wavelength? The protein studied is of highly elongated shape. Using MALS there is a procedure to correct for the shape of the protein. It remains unclear as to whether this procedure has been applied. If not, data are not valid. The molecular weights shown on the graph: are these experimentally determined values (if yes provide experimental errors) or expected elution times of a globular protein of the mass indicated?

Fig. S6E. There are no error values associated with these data. Considering that differences observed

are very modest error values are essential. The Y-axis title is "CheA activity", but no units are indicated. Again, more direct methods can be applied to study interaction.

4. Since CheW-CheR like is the novel protein and central to this story, experiments should be conducted to characterize this protein at the biochemical level; for example, does it retain the capacity to bind SAM or SAH, and does it catalyze methylation?

Based on cryo EM image it was postulated that the "middle density" is dimerized CheW-CheR like. Dimerization of CheW-CheR like was observed on native PAGE gels, however the buffer conditions used to generate these data are not available; if it were generated in the absence of reducing agent, the small amount of dimers seen could be artefactual cross linked monomers by establishing disulfide bridges among some of the three cysteine residues of the protein. Other, more precise biophysical approaches can be used to characterize the propensity of CheW-CheR like to oligomerize.

5. Efforts must be made to present in a more clear way the argumentation that the observed arrays in *Spirochaetes* support strong bending, whereas the canonical arrays do not. What are the molecular features of the novel array that are not shared by the canonical array that support extreme membrane curvature?

Reviewer #2 (Remarks to the Author):

The manuscript by Muok and colleagues reveals a new structural arrangement of the bacterial chemotactic receptor array. While the most studied arrays are C6 symmetric, the authors show that *Treponema denticola* have a P2 symmetric arrangement. The authors further suggest the location of a CheR-like domain in the structure of *T. denticola* chemoreceptor array and demonstrated conformational differences in absence of the oxygen-binding diiron protein. This is an interesting manuscript demonstrating diversity of chemoreceptor systems of bacteria.

Major points:

1. Structures do not have resolution measurements making any further structural analysis questionable. The processing should be done according to the gold standard procedure - dividing the sub-volumes into two independent half-sets, processing them independently, calculating FSC. The structures should be filtered to the resulting resolution. In the current manuscript the number of tomograms and particles are limited, therefore it is also critical to make sure that the resulting maps have isotropic resolution. Angular distribution graphs should be presented. For example, I suspect that the WT and the delta 2498 maps in figure 2 have anisotropic resolution resulting in an apparent elongation of the features.

2. The authors suggest that the unusual arrangement may be a consequence of the bacterial cell shape. This is highly intriguing, however in order to suggest that this is a general feature more experiments are needed, such as demonstrating that the other spirochetes also have such arrangement. For these experiments, spirochetes far enough on the chemosensory tree in Figure S2 should be selected.

Minor points:

3. It is hard to evaluate sequence similarity within the analyzed dataset in supplementary Figures S2 and S3, adding a scale would be necessary. Also the fonts in the figure S2 are unreadable, perhaps reducing the number of species and increasing the resolution of the image could help.

4. Structures need to be deposited to the EMDB and the accession numbers need to be stated in the text.

5. The convention for density presentation is to show density of the averages in white, the authors

should consider inverting the contrast.

6. An inconsistency in the use of terms sub-tomogram and sub-volume, only one of the two should be used.

Reviewer #3 (Remarks to the Author):

This is a very well done interdisciplinary study that combines genomic analysis, in situ cryo-ET and structural biology to describe a unique architecture of a chemoreceptor array that may have evolved to accommodate to a very high degree of membrane curvature in the very thin-bodied spirochete *Treponema denticola*. There is nothing I can find to criticize in the science, but I think that the presentation could be improved to make the work more accessible to readers.

My biggest concern is with the figures. I am not expert at looking at cryo-ET images, so the features of extra densities that are pointed out in Figures 1 and 2 were not obvious to me, in spite of the arrows. I think it would be helpful to show an example of a lattice in which these densities are not present so that a comparison could be made. I would also suggest pointing out all of these densities with their own arrows. I also comment on this in my numbered comments. Perhaps a cartoon could be added to Figure 1 in which the location of the extra densities in Td arrays and their absence in Ec arrays could be highlighted.

I also had some concerns about inadequate information in the legend and incomplete gels in figure. The Discussion is very clear, but some of the sections in the Results are hard to follow. The one on membrane curvature is especially confusing. I would suggest that the authors go through each section of the Results and make sure at all of them understand all of them. I may be wrong, but it seems that different authors wrote different sections independently.

There are a large number of Supplemental Figures some of which are quite important for the story that is being told. I suppose that space constraints make this inevitable, but it should be pointed out to the reader that it is more essential than usual to look at the Supplemental Figures.

Numbered comments with line numbers.

1)Line 27. There should be a comma after integrity, but more importantly, the statement "its loss substantially orders CheA" is too cryptic. Perhaps better would be "and in its absence, CheA becomes more ordered within the array."

2)Line 58. Insert comma after "curvature."

3)Lines 59 ff. Rewrite as Here, we used cryo-ET to examine the in vivo array structure ...(Td), which possesses the...of any bacterial species examined...thus far."

4)Line 69. Replace "to" with "in."

5)Line 70. Delete comma.

6)Lines 76-79. How many of the 'Flagellar' systems are of the F2 type? Only one? Are there other types?

7)Line 83. Rewrite as "However, not all...F2 system."

8)Line 84. Insert comma after "GTDB," and explain what GTDB is.

9)Line 85. Replace "with" with "within."

10)Lines 102-110. I realize that no methylation assays were done, but are the critical residues for binding SAM conserved in the CheRlike domain. If not, that would strengthen the argument that this domain is structural and not involved in methylation. Also, it might be mentioned here whether the F2 CheR is known to methylate the receptors.

11)Line 114. The term "location of the last common ancestor" (add "the") is vague. "location on the CheW phylogenetic tree" would be more informative. Also, I find the legend to Figure S4 inadequate. Here again, the question arises of how many F types there are. (See comment on lines 76-79.) Also, explain all of what is shown in the figure more clearly. It seems that all of the known CheW sequences are mapped.

12)Fig1B. It is certainly possible that I am just dense, but I find it very difficult to see the "densities" indicated by the blue and green arrows. It might help if all of these densities in all three strains were

indicated by arrows. It would also help to have, for comparison, a typical receptor array in which these densities were not present for comparison. The same holds for the arrangement of CheA in Figure 1C. How is CheA arranged in the typical P6 array. These points are rather subtle for the uninitiated. The same problem exists with Figure 2A and B. I presume the reader should be comparing the images in A with the one in B, but to me, they look totally different. Again, the reader should be given more help in recognizing the differences.

13)Line 142. Rewrite as "Td. However, it is unknown whether ODP...."

14)In Figure 2C, I assume that equal amounts of the reaction products were loaded, but this should be said here. Also, how significant is the difference in CheA phosphorylation; the bands are not scanned for density. Finally, how does this stimulation by the CheWCheRlike protein compare with stimulation by the normal CheW?

15)In Figure 2D, what is the left lane? Is it a MW standard? And, in both 2C and 2D, why aren't the entire gels shown instead of just slices of single lanes?

16)I found the section on CheA arrangement and array curvature very hard to follow. For example, if the array densities at layer 1 lie at an angle of 10.4° degrees to the cell axis, why are the CheA:CheW strands at an angle of 55.4° to the cell axis perpendicular to the lines in layer 1? That looks like an angular difference of 45° , not 90° . I suggest rewriting this section for clarity and perhaps putting some of the less-crucial information into Supplementary Material.

17)Line 254. Set off "in Td CheA" with commas. Same for "including Borrelia and Brachyspira" on lines 255-256.

18)Line 264. Insert comma after "breakages."

19)Line 273. Replace "produces" with "produce."

20)Line 278. Delete "a."

21)Line 280. Rewrite as "...and instead matches the helix connectivity...."

22)Line 285. Replace "cyndrillical" with "cylindrical."

23)Line 300. Insert comma after "i.e."

24)Lines 305ff. Suggest rewriting as "...area. It may also encourage...receptors. Because of this curvature, the...."

25)Lines 309ff. "...bacteria whose chemotaxis architectures have been thus far determined. The novel...arrays may better accommodate this extreme membrane curvature. We surmise...."

26)Line 314. Is Spirochaetia correct? Early in Results it was the phylum Spirochaetota.

27)Line 324. Delete comma after "array."

28)Lines 328-329. Rewrite as "...has evolved to complement the high membrane curvature and asymmetry of spirochetes."

29)Line 337. Start new paragraph.

30)Line 338. Replace "shows" with "reveal."

31)31) Lines 341 ff. Rewrite as "...the Td chemoreceptor architecture affects...in the arrays; linear CheA:CheW strands are connected only through...behavior than those that have been reported. Collectively, our results...in situ and show that ...unexpected advances to our understanding of the remarkable signaling systems of bacterial chemotaxis.

REVIEWER COMMENTS

Reviewer #1 (Remarks to the Author):

Previous work, done in large part by Prof. Briegel, has characterized by cryo-EM the chemosensory arrays of a significant number of bacteria. Here the authors have studied the chemosensory array of *Treponema denticola* and found that it was different to the so far known systems. In analogy to the canonical systems there was also a hexagonal arrangement but CheAs show a linear arrangement. Bioinformatics studies show that the system belonged to category f2 and that members of this category are basically spirochaetes. The f2 gene clusters are characterized by the presence of a novel signaling protein, namely a fusion between CheW with a CheR like domain. The densities in cryo EM images corresponding to this CheW-CheR like protein have been identified. The authors argue that this protein is the reason for the existence of this unusual arrangement and the capacity of the arrays to support extreme membrane bending, which is necessary to accommodate chemosensory arrays in bacteria with a spiral topology. This is a very interesting study combining bioinformatics, biochemistry and X-ray crystallography with cryo-EM. This study will be of impact in the scientific community since it is an important contribution in the efforts to explore the diversity of chemosensory signaling. However, this referee has a number of concerns that need to be addressed.

1. Reading this manuscript one gets the impression that it contains a large amount of data that in the end dilute the core message. In this reviewer's mind the core message is that the presence of a novel protein, the CheW-CheR like fusion, that generates hexagonal arrays that are characterized by a linear CheA arrangement, which in turn permits accommodating arrays in bacteria with a spiral topology, characterized by strong bending. The initial bioinformatic part is clear and streamlined and this referee feels that attempts could be made to streamline the remaining part of the manuscript in a similar fashion; that involves removing data that do not contribute to the key story and expanding on the novel protein, using both *in vivo* and *in vitro* approaches. For example, the authors report data from mutants in Tde2498, a previously reported oxygen sensor, and Tde2496, a protein composed of solely a MCP signaling domain. The contribution of these data to the overall story is not clear; data show that removing both proteins alters array structure, indicating that both proteins are present in the array.

Thank you for your comments and your insightful suggestions on how to improve this manuscript. We have edited the text to streamline the various sections as suggested (see marked version). The cryo-ET experiments with TDE2498 (ODP) and its cognate receptor TDE2496 are critical for this study because it is only in the respective mutant strains that the arrays are ordered to an extent where the linear arrangement of CheA is apparent (this detail is simply not visible in WT cells). The work further elucidates the effect of ODP on array structure, which complements the previous, largely *in vitro* work¹. Furthermore, because TDE2496 is a soluble receptor (i.e. no predicted transmembrane domains), it is of interest that there are no apparent cytoplasmic arrays in *Td*. These data also indicate that neither ODP nor TDE2496 is an integral component of the array. We do agree that some of the

experiments with TDE2496 can be removed (MALS and activity assays, see below) as they do not contribute to the main findings of this study.

2. Since the CheW-CheR like protein is firstly novel and secondly central to the story, more research needs to be dedicated to this protein, both in the in vivo and in vitro level. The authors use a delta CheR like mutant, but dedicate only a few lines on the cryo-EM images on page 7. A central question is whether these arrays support bending? Maybe present an image in a similar position like that of the WT in Fig. S10? Another central question is how the chemotactic behavior of the wt compares to that of the delta CheR like mutant. Are the formed arrays in the delta CheR like protein functional? Chemotaxis assays should be conducted.

Thank you very much for these insightful questions and we agree that answering them will add to the impact of our story. First, because deletion of the CheR_{like} domain significantly decreases the size of the arrays, we expect the arrays are no longer able to form an extended lattice in the curved membrane. To illustrate this point, we included an array image similar to WT in Fig. S4B as suggested. As the arrays are smaller, the axis of the Td cell is not apparent in the sub-tomogram average.

Chemotaxis assays with the CheR_{like} mutant were conducted and compared to WT cells, and there was no difference in chemotaxis between the two strains (using glucose and hemin as attractants, (Fig S6). However, this is expected as previous experiments in *E. coli* demonstrate that assembled arrays are only necessary for array cooperativity, and chemotaxis mutants that cannot assemble arrays, but are otherwise functional, remain chemotactic at wild-type levels². In this previous study, observable changes in chemotaxis activity were only apparent with sensitive FRET assays that currently are not available in the spirochete system.

3. The weakest part of this manuscript is the biochemical part. The experimental procedures for some of the experiments are lacking, such as for example the buffer and experimental conditions used for the determination of the oligomeric state (Fig. 2D), MALS (Fig. S6B, in fact SEC-MALS was used) or the experiments that led to Fig. S6F. Some more specific points are:

Thank you for pointing this out. We have now added the sections to the Methods and Materials section for the following experiments: SEC-MALS, Mass spectrometry, Isothermal Calorimetry, Chemotaxis assays, and Molecular modeling of CheR domains.

Fig. S6A: Phosphatases are frequent contaminants in recombinantly purified proteins. Control experiments are to be conducted to exclude the possibility that the reduction seen is not due to the action of contaminants. There is no clear evidence that both proteins interact. There is a wealth of more direct methods to study interaction.

As previous experiments have demonstrated that CheA is modulated by Tde2496 receptor mimetics¹, we removed these data and instead reference this published paper.

6B: AU, which wavelength? The protein studied is of highly elongated shape. Using MALS there is a procedure to correct for the shape of the protein. It remains unclear as to whether this procedure has been applied. If not, data are not valid. The molecular weights shown on the graph: are these experimentally determined values (if yes provide experimental errors) or expected elution times of a globular protein of the mass indicated?

We removed these data from the manuscript since they do not directly relate to the main conclusions of the study; however, we do note that for a particle of this size ($< \lambda/20$), the angular dependence of the light-scattering signal is small and hence the molecular weight as determined by the light-scattering and refractive index measurements is independent of particle shape.

Fig. S6E. There are no error values associated with these data. Considering that differences observed are very modest error values are essential. The Y-axis title is "CheA activity", but no units are indicated. Again, more direct methods can be applied to study interaction.

Instead of using radioisotope assays, we have now shown that the CheW-CheR_{like} protein and CheA interact by SEC-MALS. The ~220 kDa complex formed indicates a 1:1 binding ratio of CheA dimer to CheW-CheR_{like} subunit under these conditions (Figure 4B).

4. Since CheW-CheR like is the novel protein and central to this story, experiments should be conducted to characterize this protein at the biochemical level; for example, does it retain the capacity to bind SAM or SAH, and does it catalyze methylation?

We conducted native mass spectrometry (ESI-MS) of the *Td* CheW-CheR_{like} protein and the *Td* classical CheR protein and found that they both bind SAM in a 1:1 ratio (Fig. S5C). Furthermore, Isothermal Calorimetry (ITC) experiments demonstrate that the CheW-CheR_{like} protein binds SAM with an affinity of ~9 μ M (Fig. 4C). For comparison, the classical CheR protein binds with a K_d of ~20 μ M.

Although the CheR_{like} domain binds SAM with substantial affinity, we surmise that it is unlikely that it catalyzes receptor methylation for several reasons.

(1) F2 CheR_{like} does not possess the two strictly conserved residues that are essential for methyltransferase activity (R79 and Y218 in *Td* classical CheR)³. These residues are found in all classical CheR proteins across diverse phyla, including Spirochetes, and mutagenesis of these residues completely inhibits receptor methylation in other systems.

(2) The CheR-like domain does not possess two additionally conserved residues that bind to the receptor C-terminus near the receptor substrate residues (Gly152 and Val180 in *Td* classical CheR)⁴. Again, these are conserved across diverse phyla.

(3) The CheW-CheR_{like} protein interacts with CheA at the receptor tips, which would anchor it away from the adaptation region of the receptor where the adaptation sites reside.

(4) We created homology models of the *Td* classical CheR and CheR-like and mapped the F2 sequence logos onto the models (ConSurf). These models show that residues adjacent to the SAM pocket and residues on the sub-domain that binds receptors are highly conserved in the classical CheR but poorly conserved in CheR_{like}, suggesting that the two units have different interaction partners (Fig. 4D). Furthermore, on the opposing side of the SAM pocket, the CheR-like domain has higher overall conservation than the classical CheR and this interface may be involved in function.

(5) *Td* CheR_{like} possesses only a 20% identity to *Td* CheR (L-align), suggesting a different biological function for these units. Indeed, previous studies show that different CheR paralogs present in a single organism have different substrates and biological functions³. However, in these cases, the paralogs still possess the conserved catalytic residues necessary for methylation. If CheW-R_{like} does have methyltransferase activity, it acts by a different mechanism than classical CheR homologs and likely does not act on the receptor adaptation region.

Whereas the possibility that CheW-CheR_{like} has methylation activity is of interest to us, the functional relevance of such activity is not straightforward to ascertain. Currently, we do not know the receptor/protein substrates or methylation sites for CheR or CheR_{like}, we also do not have an antibody against the relevant *Td* receptors and do not know the necessary conditions to analyze these reactions. Additionally, we already know that in the absence of the CheR_{like} domain of CheW-CheR_{like}, the *Td* arrays are disassembled. Structural disassembly of the arrays in of itself could affect the ability of the receptors to act as methylation substrates. Thus, even if CheR_{like} is shown to alter methylation status, it is not trivial to distinguish whether this is a direct effect of enzymatic activity or an indirect effect that derives from the change in assembly state. For all of these reasons, we feel that pursuing the functional role of CheR_{like} methylation activity is a study onto itself and is outside of the scope of the current work, which aims to structurally characterize a new chemotaxis arrangement.

Based on cryo EM image sit was postulated that the “middle density” is dimerized CheW-CheR like. Dimerization of CheW-CheR like was observed on native PAGE gels, however the buffer conditions used to generate these data are not available; if it were generated in the absence of reducing agent, the small amount of dimers seen could be artefactual cross linked monomers by establishing disulfide bridges among some of the three cysteine residues of the protein. Other, more precise biophysical approaches can be used to characterize the propensity of CheW-cheR like to oligomerize.

To characterize the oligomeric state of the CheW-CheR_{like} protein, we conducted SEC-MALS after treating the protein with 5 mM DTT (Figure S5A). Indeed, most of the protein is present as a monomer but a small amount of dimer is present. It is worth considering that the effective concentration of the CheR_{like} domains in the arrays is very high, and hence a high affinity of the subunits may not be necessary to mediate dimerization *in vivo*. Nonetheless, we consider it an open question as to

whether CheR_{like} actually dimerizes strongly in the arrays and have modified our discussion of this point accordingly:

“The deleterious effect on array integrity with the loss of the CheR_{like} domain strongly suggests that CheR_{like} plays a key role in array assembly and stabilization, perhaps through dimerization across the ring that is encouraged through close proximity *in vivo*.”

5. Efforts must be made to present in a more clear way the argumentation that the observed arrays in Spirochaetes support strong bending, whereas the canonical arrays do not. What are the molecular features of the novel array that are not shared by the canonical array that support extreme membrane curvature?

There are four main features of these arrays not shared with the canonical array: (1) three CheA:CheW proteins comprise a strict order in the rings, including a non-canonical CheW-CheR_{like} protein (2) a linear arrangement of CheA with a preferential orientation with respect to the cell axis allows specific contacts to follow the path of least curvature, (3) an additional structural component, the CheR_{like} domain, and (4) an elongated P3 domain of CheA. We have added a paragraph to the Discussion to highlight this overall argument more clearly.

“In summary, there are several molecular features of the *Td* arrays that are not shared by the canonical system. First, *Td* has evolved three components in the CheA:CheW rings that may ensure that CheA is arranged in a strictly linear formation. This feature produces chemotaxis arrays that have a bilateral symmetry (P2), as opposed to the canonical radial symmetry (P6), and have a preferred orientation with respect to the cell axis and highly curved membrane. Second, the P2 symmetry and preferred orientation allows specific protein interaction sites to follow the path of least curvature in the cell, presumably to maintain contacts with reduced strain. Third, the arrays in *Td* include a new structural component, the CheR_{like} domain of CheW-CheR_{like}, that is crucial for maintaining array integrity. The high membrane curvature may impose substantial strain to the arrays, and the CheR_{like} domain is needed for additional stability. Lastly, the *Td* CheA protein possesses an extended P3 domain that may increase stability of the dimerization domain that would undergo additional strain from aligning to the substantially curved CheA:CheW baseplate, and/or interact with receptors that are further splayed due to the highly curved membrane. Collectively, these features support the conclusion that they have evolved to support array formation in a highly curved membrane.”

Reviewer #2 (Remarks to the Author):

The manuscript by Muok and colleagues reveals a new structural arrangement of the bacterial chemotactic receptor array. While the most studied arrays are C6 symmetric, the authors show that *Treponema denticola* have a P2 symmetric arrangement. The authors further suggest the location of a CheR-like domain in the structure of *T. denticola* chemoreceptor array and demonstrated conformational differences in absence of the oxygen-binding diiron protein. This is an interesting manuscript demonstrating diversity of chemoreceptor systems of bacteria.

Major points:

1. Structures do not have resolution measurements making any further structural analysis questionable. The processing should be done according to the gold standard procedure - dividing the sub-volumes into two independent half-sets, processing them independently, calculating FSC. The structures should be filtered to the resulting resolution. In the current manuscript the number of tomograms and particles are limited, therefore it is also critical to make sure that the resulting maps have isotropic resolution. Angular distribution graphs should be presented. For example, I suspect that the WT and the delta 2498 maps in figure 2 have anisotropic resolution resulting in an apparent elongation of the features.

Thank you for your insightful comments.

The resolution of the tomograms for the WT, $\Delta 2498$, and $\Delta 2498$ -2496 strains has been determined by the gold-standard procedure. However, due to the low number of particles available the resolution of the $\Delta\text{CheR}_{\text{like}}$ tomogram was determined by Resmap. These values have been added to Figure S4A. In this manuscript, we mainly focus on general structural differences between datasets of different strains. We believe that by using the same procedure for data processing the structures are comparable and the differences of the structures observed are reliable as all averages share similar calculated resolutions with one another.

As the reviewer requested, each average was low-passed filtered to the resolution claimed. For the WT, $\Delta 2498$, and $\Delta 2498$ -2496 strains, the filtering step did not change interpretations of the maps but reduced clarity in some regions of the maps. This is due to the fact that the reported resolution of the sub-tomogram averages only represents the average resolution of the tomogram. For instance, analysis via Resmap with the $\Delta 2498$ -2496 strain visualizes resolution across the maps and reports an average of 33 Å. But the center of the maps, where the arrays are located, are at a much higher resolution (first image below). We selected this large box size since it shows the cell axis relative to the arrays. Therefore, filtering the tomograms to the average resolution decreases visibility of regions with higher resolution, and is not appropriate in this case. However, for the ΔCheR average, filtering to the reported resolution produces maps with increased density in the center of the CheA:CheW rings, presumably due to the filtering and removal of background noise that arises from the low number of particles and poor ordering of arrays in this strain (second image below). Therefore, we have changed the manuscript to reflect this new insight and speculate that the 'middle density' is still apparent in this strain but it less defined, and there is perhaps another protein interacting in the center of the receptor hexagons.

The reviewer is correct in that the tomograms have anisotropic resolution due to the missing wedge effect that is created by averaging top and bottom views of the chemoreceptor arrays (see angular distributions of particle projections below). However, this deficiency in orthogonal views is expected and a common limitation of cryo-ET experiments of chemotaxis arrays; side-views of the arrays are often not incorporated into the averages⁵⁻⁸. This limitation does not affect the conclusions of our manuscript as we only consider general features of the arrays at angles that are not affected by the missing wedge effect (top views of arrays). Although we are interested in incorporated side-views of arrays into our averages so we can interpret the tomograms from orthogonal views, this process will be particularly complicated in this organism due to the high membrane curvature.

2. The authors suggest that the unusual arrangement may be a consequence of the bacterial cell shape. This is highly intriguing, however in order to suggest that this is a general feature more experiments are needed, such as demonstrating that the other spirochetes also have such arrangement. For these experiments, spirochetes far enough on the chemosensory tree in Figure S2 should be selected.

Although we agree that these suggested experiments are a good idea for future pursuits, we feel that they are out of the scope for this manuscript. Here, for the first time, we are able to determine the structure of arrays in a spirochete by cryo-ET, and we see that the arrangement breaks the paradigm of P6 symmetry. Our manuscript aims to investigate the components and features of these arrays to account for this atypical arrangement.

Furthermore, we have personally attempted to image another model F2 spirochete, *Borrelia burgdorferi*, for chemotaxis arrays but top and bottom views of the arrays are not apparent in any of the cells, which are essential for sub-tomogram averaging⁶. In total, we collected ~80 tomograms of the *Bb* cells and the reconstructed tomograms are available on ETDB. Other researchers have also conducted cryo-ET with spirochete cell poles (*Bb*⁹, *Treponema pallidum*¹⁰, and *Leptospira*¹¹) but top-views of the arrays have not been reported in any of these species yet. While it is unclear why top-views are present in *T. denticola* and not other spirochetes (one reason might be the presence of multiple flagellar motors in the same region where we expect the arrays to be, so they might be fragmented and are undetectable), screening several other spirochete species for array top-views

and processing these data will be extremely challenging as spirochetes are difficult to cultivate. Additionally, we are only able to distinguish the linear CheA arrangement in *T. denticola* in the ODP knock-out strains. Therefore, if one were to observe array top-views in other spirochetes, the WT cells may not reveal the CheA arrangement and further genetic experiments may be needed to be conducted to obtain such information.

Although our cryo-ET experiments in *B. burgdorferi* were unfruitful, previous genetics work (by an author of the manuscript, Chunhao Li) in this organism supports our findings on the *Td* architecture. We have added these findings to the Discussion of the manuscript:

“Bioinformatics analyses indicate that the unique protein features seen in *Td* are exclusive to all Spirochaetia (F2) systems. Indeed, gene deletion studies of *Bb* have shown that two CheW proteins, a classical CheW (CheW1) and a CheW-CheR_{like} (CheW3), are essential for array formation and chemotactic behavior, and possess unique regions at the ring interfaces¹¹. *Bb* also possesses two CheA homologs (CheA1-F8 and CheA2-F2) but only one of the homologs (CheA2) contains an elongated P3 domain and is essential for chemotaxis and pathogenicity²⁶. Therefore, we predict that a similar chemotaxis arrangement is present in *Bb*, which has a similar diameter as *Td*. However, cryo-ET experiments of *Bb* fail to produce top-view images of arrays sufficient for sub-tomogram averaging, for reasons that are unclear^{4,29}. Furthermore, cryo-ET experiments of cell poles in other spirochetes have been conducted (*Treponema pallidum*³⁰ and *Leptospira*³¹) but top-views of arrays in these species have not been reported yet.”

Minor points:

3. It is hard to evaluate sequence similarity within the analyzed dataset in supplementary Figures S2 and S3, adding a scale would be necessary. Also the fonts in the figure S2 are unreadable, perhaps reducing the number of species and increasing the resolution of the image could help.

Thank you for these comments. We have now added a scale to both figures and increased the size of the sequence logos. We have also submitted these figures as separate high-resolution PDFs instead of in-text images. We have also now included in the main manuscript, an additional figure (Figure 2) to summarize the data in the former Figure S2.

4. Structures need to be deposited to the EMDB and the accession numbers need to be stated in the text.

The EMDB accession numbers are now added to the body of the text.

5. The convention for density presentation is to show density of the averages in white, the authors should consider inverting the contrast.

Thank you for pointing this out. It is indeed standard for several cryo-EM fields to present density inverted to what we use here. However, for cryo-ET studies it is common for the protein density to be shown in black (All our previous work on the

arrays has been presented this way. Since the chemotaxis community is familiar with this presentation style, we prefer to keep the averages as they are).

6. An inconsistency in the use of terms sub-tomogram and sub-volume, only one of the two should be used.

Thank you for pointing this out to us. We have now only use sub-tomogram averaging in the text.

Reviewer #3 (Remarks to the Author):

This is a very well done interdisciplinary study that combines genomic analysis, in situ cryo-ET and structural biology to describe a unique architecture of a chemoreceptor array that may have evolved to accommodate to a very high degree of membrane curvature in the very thin-bodied spirochete *Treponema denticola*. There is nothing I can find to criticize in the science, but I think that the presentation could be improved to make the work more accessible to readers.

My biggest concern is with the figures. I am not expert at looking at cryo-ET images, so the features of extra densities that are pointed out in Figures 1 and 2 were not obvious to me, in spite of the arrows. I think it would be helpful to show an example of a lattice in which these densities are not present so that a comparison could be made. I would also suggest pointing out all of these densities with their own arrows. I also comment on this in my numbered comments. Perhaps a cartoon could be added to Figure 1 in which the location of the extra densities in *Td* arrays and their absence in *Ec* arrays could be highlighted.

I also had some concerns about inadequate information in the legend and incomplete gels in figure.

The Discussion is very clear, but some of the sections in the Results are hard to follow. The one on membrane curvature is especially confusing. I would suggest that the authors go through each section of the Results and make sure at all of them understand all of them. I may be wrong, but it seems that different authors wrote different sections independently.

There are a large number of Supplemental Figures some of which are quite important for the story that is being told. I suppose that space constraints make this inevitable, but it should be pointed out to the reader that it is more essential than usual to look at the Supplemental Figures.

Thank you for your helpful comments on the improvement of this manuscript. We have now gone over each of the results sections for improvement, focusing especially the section about membrane curvature. Also, we have now added an additional main figure (Figure 1) which shows the canonical chemotaxis array arrangement, including densities of Layer 1 and Layer 2 in *E. coli*. We have also added additional, and larger arrows to the figures showing the density features in *Td*. To reduce the number of figures in the supplemental section, we have moved 2 figures and 1 table from the supplemental to the main text, and removed a supplemental figure (the former Fig. S7) as we felt it was redundant with other figures that illustrate the CheA:CheW ring arrangement.

Numbered comments with line numbers.

1)Line 27. There should be a comma after integrity, but more importantly, the statement “its loss substantially orders CheA” is too cryptic. Perhaps better would be “and in its absence, CheA becomes more ordered within the array.”

Done

2)Line 58. Insert comma after “curvature.”

Done

3)Lines 59 ff. Rewrite as Here, we used cryo-ET to examine the in vivo array structure ...(Td), which possesses the...of any bacterial species examined...thus far.”

Done

4)Line 69. Replace “to” with “in.”

Done

5)Line 70. Delete comma.

Done

6)Lines 76-79. How many of the ‘Flagellar’ systems are of the F2 type? Only one? Are there other types?

We are sorry for the confusion. There are 17 classes of chemotaxis system originally predicted to control flagellar motility, thus the name "Flagellar" and the letter F. Although the obligate link to flagella-mediated motility no longer holds, the name of the classes remains. We have now stated this more clearly in the bioinformatics section line 7.

The question of how many systems are of the type F2 is based on the genomes curated in MiST3 database. We can confidently mention the 306 CheA-F2 present in MiST3 and we changed the Methods and Materials section to reflect this on line 838.

7)Line 83. Rewrite as “However, not all...F2 system.”

Done

8)Line 84. Insert comma after “GTDB,” and explain what GTDB is.

We have now included on line 107: "There are 1096 genomes assigned to the Spirochaetota phylum in the Genome Taxonomy Database (GTDB) [16], a microbial taxonomy based on genome phylogeny..."

9)Line 85. Replace “with” with “within.”

Done

10)Lines 102-110. I realize that no methylation assays were done, but are the critical residues for binding SAM conserved in the CheRlike domain. If not, that would strengthen the argument that this domain is structural and not involved in

methylation. Also, it might be mentioned here whether the F2 CheR is known to methylate the receptors.

Thank you for these comments. *Td* does possess homologs of the adaptation proteins CheR and CheB and they are on the same operon¹², but it has not been experimentally confirmed that the *Td* CheR protein methylates receptors or that *Td* CheB catalyzes demethylation. We have experimentally determined that both the *Td* classical CheR and the CheR_{like} domain bind SAM (Fig. 4C). However, CheR_{like} does not possess the strictly conserved residues for methyltransferase activity, or the strictly conserved residues for receptor binding. The classical CheR in *Td* does possess these residues. Therefore, we surmise that CheR_{like} has a different biological function than the classical CheR and if it is capable of methylation, would utilize a different mechanism than classical CheR homologs. These insights have been added to the manuscript. See also the response to reviewer 1 (question 4) on this point.

11)Line 114. The term "location of the last common ancestor" (add "the") is vague. "location on the CheW phylogenetic tree" would be more informative. Also, I find the legend to Figure S4 inadequate. Here again, the question arises of how many F types there are. (See comment on lines 76-79.) Also, explain all of what is shown in the figure more clearly. It seems that all of the known CheW sequences are mapped. Thank you for calling our attention to these ambiguities. We made the following changes below to clarify our manuscript.

For the sake of accuracy, we changed the segment:

"F2 systems contain three proteins with a CheW domain: the classical scaffold CheW, CheW-CheRlike, and the P5 domain from the histidine kinase CheA."

to:

"F2 systems contain three proteins with a CheW domain: the classical scaffold CheW, CheW-CheRlike, and the histidine kinase CheA."

We also decided to remove the sentence mentioned by the reviewer:

"Phylogenetic mapping of CheW-F2 proteins indicate the location of last common ancestor (Fig. S4, see 115 Supplementary methods)"

This information is not necessary in the main text as the identification of the last common ancestor of the CheW-F2 among other CheW proteins in these organisms is simply an accessory analysis to identify CheW from the F2 class. This is a necessary step as there is no hidden Markov model to classify CheW into chemotaxis classes. This procedure was better explained in the Methods section "Classification of CheW".

We changed the header "Comparison of the CheW domains in CheA, CheW-CheRlike and CheW" to:

"Comparison of the CheW domains in CheA-F2, CheW-CheRlike and CheW-F2"

For coherence, we also changed the main text segment:

"To investigate sequence patterns of the three CheW domains, we analyzed non-redundant sequence datasets of CheW, CheW-CheRlike, and CheA-F2 from all 117 genomes with at least one CheA-F2. The final alignments for each group contain the CheW domain portion of 74 CheW proteins, 59 CheW-CheRlike proteins, and 73 CheA proteins."

to

"To investigate sequence patterns of the three CheW domains, we analyzed non-redundant sequence datasets of CheW-F2, CheW-CheRlike, and CheA-F2 from all 117 genomes with at least one CheA-F2. The final alignments for each group contain the CheW domain portion of 74 CheW-F2 proteins, 59 CheW-CheRlike proteins, and 73 CheA-F2 proteins."

To improve clarity, we changed the legend of Figure S4 from:

"Phylogenetic tree of CheW sequences in genomes with at least one CheA-F2 suggests a last common ancestor of CheW-F2 sequences. We mapped the sequences of CheW from the classes F2 (red), F5(purple), F7 (light green), F8 (green) and ACF (blue). The larger red internal node marks a candidate of the last common ancestor of CheW-F2."

to

"Phylogenetic tree of a non-redundant set of CheW protein sequences in genomes with at least one CheA-F2. The initial classification of CheW classes are mapped to the tree nodes: F2 (red), F5(purple), F7 (light green), F8 (green) and ACF (blue). The clustering of the CheW classified as class F2 suggests a last common ancestor of CheW-F2 sequences (larger red internal node). This clustering allows us to extrapolate the conservative selection of F2 sequences to other CheW proteins in the cluster to get a more inclusive set."

12)Fig1B. It is certainly possible that I am just dense, but I find it very difficult to see the "densities" indicated by the blue and green arrows. It might help if all of these densities in all three strains were indicated by arrows. It would also help to have, for comparison, a typical receptor array in which these densities were not present for comparison. The same holds for the arrangement of CheA in Figure 1C. How is CheA arranged in the typical P6 array. These points are rather subtle for the uninitiated. The same problem exists with Figure 2A and B. I presume the reader should be comparing the images in A with the one in B, but to me, they look totally different. Again, the reader should be given more help in recognizing the differences.

Thank you for your suggestions, we have now changed the manuscript to more accurately point to density features. We have added a new figure, Figure 1, in which

the typical P6 arrangement is illustrated. We have also added more, and larger arrows to Figure 3 (formerly Figure 1) and labeled the 'middle density' feature in Figure 4 (formerly Figure 2) to help illustrate the features being examined.

13)Line 142. Rewrite as "Td. However, it is unknown whether ODP...."

Done

14)In Figure 2C, I assume that equal amounts of the reaction products were loaded, but this should be said here. Also, how significant is the difference in CheA phosphorylation; the bands are not scanned for density. Finally, how does this stimulation by the CheW-CheRlike protein compare with stimulation by the normal CheW?

This data has been replaced. At the request of Reviewer 1, we have now determined CheW-R_{like} dimerization and interactions with CheA by MALS (Fig. 4B). These results demonstrate a minor dimer component and a 1:1 interaction with a dimeric CheA.

15)In Figure 2D, what is the left lane? Is it a MW standard? And, in both 2C and 2D, why aren't the entire gels shown instead of just slices of single lanes?

At the request of Reviewer 1, this figure was removed and replaced with MALS data. This new data also demonstrates a minor dimer component of CheW-CheR_{like} (Fig. S5A).

16)I found the section on CheA arrangement and array curvature very hard to follow. For example, if the array densities at layer 1 lie at an angle of 10.4° degrees to the cell axis, why are the CheA:CheW strands at an angle of 55.4° to the cell axis perpendicular to the lines in layer 1? That looks like an angular difference of 45°, not 90°. I suggest rewriting this section for clarity and perhaps putting some of the less-crucial information into Supplementary Material.

Thank you for your recommendation, we have rewritten and shortened this section for clarity. We are trying to establish that the densities at Layer 1 produce apparent lines in the tomograms that run relatively parallel to the cell axis. This formation puts the 'strands' of connected CheA:CheW rings relatively perpendicular to the cell axis. Therefore, the CheA:CheW rings have to bend to follow the cell curvature. To make this more apparent we have now added additional labeling to Figure S8A.

17)Line 254. Set off "in Td CheA" with commas. Same for "including Borrelia and Brachyspira" on lines 255-256.

Done

18)Line 264. Insert comma after "breakages."

Done

19)Line 273. Replace "produces" with "produce."

Done

20)Line 278. Delete "a."

Done

21)Line 280. Rewrite as "...and instead matches the helix connectivity...."

Done

22)Line 285. Replace "cyndrillical" with "cylindrical."

Done

23)Line 300. Insert comma after "i.e."

Done

24)Lines 305ff. Suggest rewriting as "...area. It may also encourage...receptors. Because of this curvature, the...."

Done

25)Lines 309ff. "...bacteria whose chemotaxis architectures have been thus far determined. The novel...arrays may better accommodate this extreme membrane curvature. We surmise...."

Done

26)Line 314. Is Spirochaetia correct? Early in Results it was the phylum Spirochaetota.

We apologize for the confusion. The reviewer is correct in that "Spirochaetia F2" is redundant, as all Spirochaetia have an F2 system. We have now changed the text to say "all Spirochaetia (F2) systems."

27)Line 324. Delete comma after "array."

Done

28)Lines 328-329. Rewrite as "...has evolved to complement the high membrane curvature and asymmetry of spirochetes."

Done

29)Line 337. Start new paragraph.

30)Line 338. Replace "shows" with "reveal."

Done

31)31) Lines 341 ff. Rewrite as "...the Td chemoreceptor architecture affects...in the arrays; linear CheA:CheW strands are connected only through...behavior than those that have been reported. Collectively, our results...in situ and show that ...unexpected advances to our understanding of the remarkable signaling systems of bacterial chemotaxis.

Done

Literature Cited:

1. Muok, A. R., Deng, Y., Gumerov, V. M., Chong, J. E., DeRosa, J. R., Kurniyati, K., Coleman, R. E., Lancaster, K. M., Li, C., Zhulin, I. B. & Crane, B. R. A di-iron protein

- recruited as an Fe[II] and oxygen sensor for bacterial chemotaxis functions by stabilizing an iron-peroxy species. *Proc. Natl. Acad. Sci.* **116**, 14955–14960 (2019).
2. Piñas, G. E., Frank, V., Vaknin, A. & Parkinson, J. S. The source of high signal cooperativity in bacterial chemosensory arrays. *Proc. Natl. Acad. Sci.* **113**, 3335–3340 (2016).
 3. García-Fontana, C., Reyes-Darias, J. A., Muñoz-Martínez, F., Alfonso, C., Morel, B., Ramos, J. L. & Krell, T. High specificity in CheR methyltransferase function: CheR2 of *Pseudomonas putida* is essential for chemotaxis, whereas CheR1 is involved in biofilm formation. *J. Biol. Chem.* **288**, 18987–18999 (2013).
 4. Djordjevic, S. & Stock, M. A. Chemotaxis receptor recognition by protein methyltransferase CheR. *Nat. Struct. Biol.* **5**, 446–450 (1998).
 5. Briegel, A., Ames, P., Gumbart, J., Oikonomou, C., Parkinson, J. S. & Jensen, G. J. The mobility of two kinase domains in the *Escherichia coli* chemoreceptor array varies with signaling state. *Ariane*. **14**, 384–399 (2010).
 6. Briegel, A., Ortega, D. R., Tocheva, E. I., Wuichet, K., Zhuo, L., Songye, C., Müller, A., Iancu, C. V., Murphy, G. E., Dobro, M. J., Zhulin, I. B. & Jensen, G. J. Universal architecture of bacterial chemoreceptor arrays. *Proc. Natl. Acad. Sci. U. S. A.* **106**, 17181–17186 (2009).
 7. Briegel, A., Ding, H. J., Li, Z., Werner, J., Gitai, Z., Dias, D. P., Jensen, R. B. & Jensen, G. J. Location and architecture of the *Caulobacter crescentus* chemoreceptor array. *Mol. Microbiol.* **69**, 30–41 (2008).
 8. Briegel, A., Li, X., Bilwes, a. M., Hughes, K. T., Jensen, G. J. & Crane, B. R. Bacterial chemoreceptor arrays are hexagonally packed trimers of receptor dimers networked by rings of kinase and coupling proteins. *Proc. Natl. Acad. Sci.* **109**, 3766–3771 (2012).
 9. Xu, H., Raddi, G., Liu, J., Charon, N. W. & Li, C. Chemoreceptors and flagellar motors are subterminally located in close proximity at the two cell poles in spirochetes. *J. Bacteriol.* **193**, 2652–2656 (2011).
 10. Liu, J., Howell, J. K., Bradley, S. D., Zheng, Y., Zhou, Z. H. & Norris, S. J. Cellular Architecture of *Treponema pallidum*: Novel Flagellum, Periplasmic Cone, and Cell Envelope as Revealed by Cryo Electron Tomography. *J. Mol. Biol.* **403**, 546–561 (2010).
 11. Raddi, G., Morado, D. R., Yan, J., Haake, D. A., Yang, X. F. & Liu, J. Three-dimensional structures of pathogenic and saprophytic *Leptospira* species revealed by cryo-electron tomography. *J. Bacteriol.* **194**, 1299–1306 (2012).
 12. Sim, J. H., Shi, W. & Lux, R. Protein-protein interactions in the chemotaxis signalling pathway of *Treponema denticola*. *Microbiology* **151**, 1801–1807 (2005).

REVIEWER COMMENTS

Reviewer #1 (Remarks to the Author):

This version of the manuscript has been significantly improved as compared to the initial submission. A number of non-essential data has been removed and other data replaced by more potent approaches like native mass spectrometry or microcalorimetry. Additional experiments such as the chemotaxis assays provide interesting additional information on the new CheWR signaling protein. The manuscript has been streamlined and the novelty of this study is now clearly distilled in the Discussion. This referee has only a number of relatively minor comments/concerns that should be addressed.

Line 83: full stop after MIST3.

Fig. 4B: Complex formation has been shown using SEC-MALS. To verify complex formation, has the fraction corresponding to the peak at 220 kDa been analysed by SDS-PAGE? Two proteins should be visible on this gel.

Fig. 4C: Replace "Integrated titration data from ITC experiments determine that CheW-R like binds SAM with slightly higher affinity than the classical Td CheR." with something like "Thermodynamic binding parameters derived from the microcalorimetric titration of CheW-Rlike and CheR with SAM..."

Fig. S5: Add m/z to the x-axis, add the sequence derived mass of both proteins to the legend

Signed: Tino Krell

Reviewer #2 (Remarks to the Author):

The manuscript "Atypical chemoreceptor arrays accommodate high membrane curvature" by Muok et al from the Briegel group has been previously reviewed by me in Nature Communications and has improved over the previous version. I am satisfied with the way the second comment and the minor comments were addressed.

However, the handling of structures is still unsatisfactory and is below the standards of the cryo-EM community. While the authors have provided the resolution values, it is also compulsory to provide the FSC curves. In the current case the structures have both local resolution variations as well as anisotropic resolution in different directions of the maps. Figure 4E still seems to have the signs of overfitting.

The authors attempt to not over interpret the maps, however as the structures are reported and potentially could be used in other research, therefore local and anisotropic resolution should be evaluated for all the maps. Maps should be filtered to local resolution. For the description of the anisotropy of resolution the angular distribution graphs provided in the reply to the reviews could be used.

Useful tools to deal with local resolution are `reliion_postprocess` with the `-locres` option, which requires half-maps or LocalDeblur (<https://doi.org/10.1093/bioinformatics/btz671>) which does not require half-maps. Other tools could be used as well.

3DFSC server (<https://doi.org/10.1038/nmeth.4347>) produces FSC curves including the directionality via a web interface.

Additional comments:

1. The authors added figure 1 as an introductory. It is not clear from which citation exactly it comes from; it should be stated in the figure legend as well. An EMDB identifier should be stated if available. Most importantly, however is that panel D has signs of overfitting which is noise, apparent corners of the box in which sub-tomogram averaging was performed. If the authors have the half-maps from which the structure has been generated, they should filter it to local resolution, if not – I suggest using LocalDeblur (<https://doi.org/10.1093/bioinformatics/btz671>) or alternatives.

2. (minor) The authors call resulting averages "averaged tomograms" (i.e. line 295), this is a non-conventional name, usually "average", "map", "sub-tomogram average" are used, I suggest revising.

3. The authors should use high-resolution images for the panels, red and yellow crosses in the middle of panels Fig 1C, Fig 3A,C, Fig. S8B suggest that the panels have been produced as snapshots from Imod; this limits the display image quality.

Reviewer #3 (Remarks to the Author):

This revised version of the manuscript is far more readable. All of my major concerns have been dealt with. If the other reviewers are satisfied with the extensive response of the authors to their comments, then only the minor editorial changes listed below should be made.

- 1) Line 57. Replace "their" with "its."
- 2) Line 58. Rewrite as "...in TD, which is likely caused by the high curvature...."
- 3) Line 59. Replace "Genetics" with "Genetic."
- 4) Line 60. Replace "reveals" with "reveal."
- 5) Line 66. Replace "assumed" with "realized."
- 6) Line 83. Insert comma after "species," and insert period after "MiST3."
- 7) Line 88. Replace "appears" with "appear."
- 8) Line 100. Replace "only found" with "found only."
- 9) Line 108. Insert comma after "results."
- 10) Lines 124-125. Rewrite as "In top views...receptors), membrane-associated...."
- 11) Line 165. Space after "genome."
- 12) Line 169. The "S" on "S-adenosylmethionine" should be capitalized."
- 13) Line 173. The "C" on "C-terminal" should be capitalized.
- 14) Line 177. Insert commas after "1AF7)."
- 15) Line 188. Replace "dimished" with "diminished" and insert comma after "density."
- 16) Line 191. Rewrite as "However, this result is expected, as previous...."
- 17) Line 193. Replace "remain chemotactic" with "support chemotaxis."
- 18) Line 197. Replace "assume the location of" with "correspond to."
- 19) Line 202. Insert comma after "i.e."
- 20) Line 225. Insert comma after "...cells)"
- 21) Line 230. Insert "other" before "organisms."
- 22) Line 231. Rewrite as "...for the arrangement of their chemotaxis arrays thus far."
- 23) Line 266. Replace "Differing" with "Different."
- 24) Line 277. Replace "been thus far" with "thus far been."
- 25) Line 294. Delete "these."
- 26) Line 313. Replace "is" with "may be."
- 27) Line 324. Rewrite as "and that they possess."
- 28) Line 325. Inset comma after "F2)."
- 29) Line 331. Insert comma after "Leptospira)."
- 30) Line 346. Rewrite as "a novel arrangement of the chemotaxis array that has evolved to complement [not compliment] the high membrane curvature...."
- 31) Line 354. Rewrite as "...cryo-ET; this method reconstitutes...."

REVIEWER COMMENTS

Reviewer #1 (Remarks to the Author):

This version of the manuscript has been significantly improved as compared to the initial submission. A number of non-essential data has been removed and other data replaced by more potent approaches like native mass spectrometry or microcalorimetry. Additional experiments such as the chemotaxis assays provide interesting additional information on the new CheWR signaling protein. The manuscript has been streamlined and the novelty of this study is now clearly distilled in the Discussion. This referee has only a number of relatively minor comments/concerns that should be addressed.

Thank you for the suggestions on the improvement of this manuscript. We think the additional biochemical experiments complements our structural data and increases the overall impact of this work.

Line 83: full stop after MIST3.

Done

Fig. 4B: Complex formation has been shown using SEC-MALS. To verify complex formation, has the fraction corresponding to the peak at 220 kDa been analysed by SDS-PAGE? Two proteins should be visible on this gel.

Thank you for your suggestion on this experiment. As the SEC-MALS system (both at Leiden and Cornell) is not equipped with a fractionator to collect eluted volumes, we conducted this experiment using a standard SEC and ran the corresponding elution peaks on SDS-PAGE to show the presence of both proteins. The corresponding lane was added to Figure 4B. The full gel image is included in the source data file.

Fig. 4C: Replace "Integrated titration data from ITC experiments determine that CheW-R like binds SAM with slightly higher affinity than the classical Td CheR." with something like "Thermodynamic binding parameters derived from the microcalorimetric titration of CheW-Rlike and CheR with SAM..."

Done

Fig. S5: Add m/z to the x-axis, add the sequence derived mass of both proteins to the legend.

Done

Signed: Tino Krell

Reviewer #2 (Remarks to the Author):

The manuscript "Atypical chemoreceptor arrays accommodate high membrane curvature" by Muok et al from the Briegel group has been previously reviewed by me in Nature Communications and has improved over the previous version. I am satisfied with the way the second comment and the minor comments were addressed.

However, the handling of structures is still unsatisfactory and is below the standards of the cryo-EM community. While the authors have provided the resolution values, it is also compulsory to provide the FSC curves. In the current case the structures have both local resolution variations as well as anisotropic resolution in different directions of the maps. Figure 4E still seems to have the signs of overfitting.

The authors attempt to not over interpret the maps, however as the structures are reported and potentially could be used in other research, therefore local and anisotropic resolution should be evaluated for all the maps. Maps should be filtered to local resolution. For the description of the anisotropy of resolution the angular distribution graphs provided in the reply to the reviews could be used.

Thank you for bringing these matters to our attention. We have now filtered the maps to local resolution using Relion. For each map, we have calculated the resolution in a masked region that includes the central receptor hexagon and CheA:CheW ring (Fig. S4A). We have now included the FSC curves for the maps, both masked and unmasked (Figures S16A-D). We have also included the angular distribution graphs provided in the

previous reply (Figures S15A-D). All figures have been replaced with the filtered maps. To account for these additional post-processing steps, we have expanded our Methods and Materials section for the Cryo-ET section. We have added:

“Tomograms were reconstructed using simultaneous iterative reconstruction with iteration number set to 6 and binning set to 2. The resulting pixel size of the tomograms was 7.026 Å (for WT, $\Delta 2498$, and $\Delta 2498\Delta 2986$ strains) and 6.54 Å (for the $\Delta \text{CheR}_{\text{like}}$ strain). Dynamo was used for manual particle picking and sub-tomogram averaging^{43,44}. As only top- and bottom- views of the arrays could be used for sub-tomogram averaging, the resulting maps are anisotropic in resolution (Fig. S15A-D). Half maps were generated by the ‘gold standard procedure’ on Dynamo. FSC calculations and local resolution filtering were done on Relion (Fig. S16A-D). The reported resolution was calculated at FSC = 0.3 within a masked region that contains the center receptor hexagon and one CheA:CheW ring (Fig. S4A). The size of the mask was 30 px X 30 px X 40 px with a soft edge of 10 px.”

Useful tools to deal with local resolution are `relion_postprocess` with the `-locres` option, which requires half-maps or `LocalDeblur` (<https://doi.org/10.1093/bioinformatics/btz671>) which does not require half-maps. Other tools could be used as well. 3DFSC server (<https://doi.org/10.1038/nmeth.4347>) produces FSC curves including the directionality via a web interface.

Additional comments:

1. The authors added figure 1 as an introductory. It is not clear from which citation exactly it comes from; it should be stated in the figure legend as well. An EMDB identifier should be stated if available. Most importantly, however is that panel D has signs of overfitting which is noise, apparent corners of the box in which sub-tomogram averaging was performed. If the authors have the half-maps from which the structure has been generated, they should filter it to local resolution, if not – I suggest using `LocalDeblur` (<https://doi.org/10.1093/bioinformatics/btz671>) or alternatives.

Thank you for bringing this fact to our attention. We have now changed these images to maps that are currently deposited in the EMDB and referenced it appropriately. Nevertheless, they still illustrate the arrangement of a ‘canonical’ P6 chemotaxis array.

2. (minor) The authors call resulting averages “averaged tomograms” (i.e.

line 295), this is a non-conventional name, usually "average", "map", "sub-tomogram average" are used, I suggest revising.

This has now been changed to "sub-tomogram averages".

3. The authors should use high-resolution images for the panels, red and yellow crosses in the middle of panels Fig 1C, Fig 3A,C, Fig. S8B suggest that the panels have been produced as snapshots from Imod; this limits the display image quality.

These images have now been updated.

Reviewer #3 (Remarks to the Author):

This revised version of the manuscript is far more readable. All of my major concerns have been dealt with. If the other reviewers are satisfied with the extensive response of the authors to their comments, then only the minor editorial changes listed below should be made.

Thank you very much for your careful editing with this manuscript. Your comments and editorial changes have greatly improved the message of our findings.

1) Line 57. Replace "their" with "its."

Done

2) Line 58. Rewrite as "...in TD, which is likely caused by the high curvature...."

Done

3) Line 59. Replace "Genetics" with "Genetic."

Done

4) Line 60. Replace "reveals" with "reveal."

Done

5) Line 66. Replace "assumed" with "realized."

Done

6) Line 83. Insert comma after "species," and insert period after "MiST3."

Done

7) Line 88. Replace "appears" with "appear."

Done

8) Line 100. Replace "only found" with "found only."

Done

9) Line 108. Insert comma after "results."

Done

10) Lines 124-125. Rewrite as "In top views...receptors), membrane-associated...."

Done

11) Line 165. Space after "genome."

Done

12) Line 169. The "S" on "S-adenosylmethionine" should be capitalized."

Done

13) Line 173. The "C" on "C-terminal" should be capitalized.

Done

14) Line 177. Insert commas after "1AF7)."

Done

15) Line 188. Replace "dimished" with "diminished" and insert comma after "density."

Done

16) Line 191. Rewrite as "However, this result is expected, as previous...."

Done

17) Line 193. Replace "remain chemotactic" with "support chemotaxis."

Done

18) Line 197. Replace "assume the location of" with "correspond to."

Done

19) Line 202. Insert comma after "i.e."

Done

20) Line 225. Insert comma after "...cells)"

Done

21) Line 230. Insert "other" before "organisms."

Done

22) Line 231. Rewrite as "...for the arrangement of their chemotaxis arrays thus far."

Done

23) Line 266. Replace "Differing" with "Different."

Done

24) Line 277. Replace "been thus far" with "thus far been."

Done

25) Line 294. Delete "these."

Done

26) Line 313. Replace "is" with "may be."

Done

27) Line 324. Rewrite as "and that they possess."

Done

28) Line 325. Inset comma after "F2)."

Done

29) Line 331. Insert comma after "Leptospira)."

Done

30) Line 346. Rewrite as "a novel arrangement of the chemotaxis array that has evolved to complement [not compliment] the high membrane curvature...."

Done

31) Line 354. Rewrite as "...cryo-ET; this method reconstitutes...."

Done

REVIEWERS' COMMENTS

Reviewer #2 (Remarks to the Author):

The original reviewer 2 is happy with the updated version of the manuscript and congratulates the authors on the exciting work.